# Molecular characterization of organic aerosols in the Kathmandu Valley, Nepal: insights into primary and secondary sources

Xin Wan[1],[8] Shichang Kang[2,7], Maheswar Rupakheti[3,4], Qianggong Zhang[1,7], Lekhendra Tripathee[2], Junming Guo[2], Pengfei Chen[2], Dipesh Rupakheti[2], Arnico K. Panday[5], Mark G. Lawrence[3], Kimitaka Kawamura[6], Zhiyuan Cong[1,7*]

[1] Key Laboratory of Tibetan Environment Changes and Land Surface Processes, Institute of Tibetan Plateau Research, Chinese Academy of Sciences (CAS), Beijing 100101, China

[2] State Key Laboratory of Cryospheric Science, Northwest Institute of Eco-Environment and Resources, CAS, Lanzhou 730000, China

[3] Institute for Advanced Sustainability Studies (IASS), Potsdam 14467, Germany

[4] Himalayan Sustainability Institute (HIMSI), Kathmandu, Nepal

[5] International Centre for Integrated Mountain Development (ICIMOD), Khumaltar, Lalitpur, Nepal

[6] Chubu Institute for Advanced Studies, Chubu University, Kasugai 487-8501, Japan

[7] CAS Center for Excellence in Tibetan Plateau Earth Sciences, Beijing 100101, China

[8] University of Chinese Academy of Sciences, Beijing 100039, China

*Corresponding author:

E-mail address: zhiyuancong@itpcas.ac.cn

**Abstract:**

Organic atmospheric aerosols in the Hindu Kush-Himalayan-Tibetan Plateau region are still poorly characterized. To better understand the sources and formation processes of the primary organic aerosols (POA) and secondary organic aerosol (SOA) in the foothills region of the central Himalaya, we studied atmospheric aerosol samples collected over a one-year period from April 2013 to April 2014 at the suburban site of Bode in the Kathmandu Valley. We measured major ions, organic carbon (OC), elemental carbon (EC), and various organic tracers emitted by specific sources. Tracer-based estimation methods were employed to characterize aerosol species, identify their likely sources, and apportion contributions from each source. The concentrations of OC and EC increased during winter with a maximum monthly average in January. Levoglucosan (an organic tracer for biomass burning), OC, and EC showed similar seasonal variations throughout the year. With an annual average concentration of $788\pm685$ ng m$^{-3}$ (ranging from 58.8 to 3079 ng m$^{-3}$), levoglucosan was observed as the dominant species among all the analyzed organic tracers. Biomass burning contributed a significant fraction to OC, averaging $24.9\pm10.4\%$ during the whole year, and up to $36.3\pm10.4\%$ in the post-monsoon season. On an annual average basis, anthropogenic toluene-derived secondary OC accounted for 8.8% and biogenic secondary OC contributed 6.2% to total OC. The annual contribution of fugal-spores to OC was 3.2% with the maximum during the monsoon (5.9%). For plant debris, it accounted for 1.4% of OC during the monsoon. Therefore, OC is mainly associated with biomass burning and other anthropogenic activity in the Kathmandu Valley. Our findings are conducive to designing control measures to mitigate the heavy air pollution and its impacts in the Kathmandu Valley and surrounding area.

# 1. Introduction

The Indo-Gangetic Plain (IGP) region in South Asia, is a global air pollution hotspot. Atmospheric pollutants [e.g., organic carbon (OC), black carbon (BC) and $SO_2$] from South Asia have been increasing in recent decades (Ramanathan et al., 2005;Muzzini and Aparicio, 2013;Lawrence and Lelieveld, 2010). While these pollutants are of concern locally near the emission sources, they can also, in short span of time, be transported to rural and remote regions over a long distance. This results in an annually recurring regional scale haze, referred to as atmospheric brown clouds (ABC), that covers a large area from the Himalayan range to the Indian Ocean (Ramanathan et al., 2007). Until recently the emissions, types, levels, atmospheric transport and transformation, impacts and mitigation of various atmospheric pollutants were not well characterized in the vast mountain areas and the foothills region in South Asia. In this context, the international project of "A Sustainable Atmosphere for the Kathmandu Valley (SusKat)" was launched, aiming to comprehensively understand the causes of the severe air pollution in the region, and identifying appropriate solutions to reduce its impacts (Rupakheti et al., 2018). This paper presents analyses of samples collected as part of the SusKat field campaign.

The Kathmandu Valley, the capital region of Nepal, is considered as one of the most polluted cities in South Asia and the largest metropolitan region in the foothills of the Hindu Kush-Himalayas-Tibetan Plateau (HKHTP) region, facing rapid but unplanned urbanization, with the current population of approximately 4 million (Muzzini and Aparicio, 2013). Additionally, due to the bowl-shaped topography, the free flow of air is restricted, resulting in poor air quality (Panday and Prinn, 2009;Shakya et al., 2010;Mues et al., 2017). Giri et al. (2006) showed that the $PM_{10}$ concentrations in Kathmandu were about 2-4 times higher than the guidelines prescribed by the World Health Organization (WHO) (50 µg m$^{-3}$ of 24-hour mean) (WHO, 2006). More recently, Shakya et al. (2017) reported that daily average mass concentrations of $PM_{2.5}$ at seven locations in the Kathmandu Valley during 2014 were about 5 times higher than the WHO guidelines (25 µg m$^{-3}$ of 24-hour mean) (WHO, 2006). Beside particulate matter, recent studies have pointed out that ground-level ozone ($O_3$) is also of concern (Mahata et al., 2017b;Bhardwaj et al., 2017). Putero et al. (2015) found $O_3$ levels at Pakanajol in the city center to exceed the WHO's 8-hour maximum ozone guidelines of 100 µg m$^{-3}$ on 125 days in a year, while Mahata et al. (2017b) reported such exceedance was for nearly 3 months at Bode (where sampling for this study was

conducted) and 6 months at Nagarkot, a hilltop site downwind of the Kathmandu Valley. The concentrations of acetonitrile and isoprene (precursor for both $O_3$ and secondary organic aerosol, SOA) investigated by Sarkar et al. (2016) in the Kathmandu Valley were comparable with the highest reported elsewhere in the world. Air pollution is a clear threat to human health (leading to respiratory disease, cardiovascular disease, cancer, etc.), agricultural productivity and revenues from tourism in the Kathmandu Valley and surrounding regions (Putero et al., 2015;Shakya et al., 2016).

Carbonaceous aerosols [organic carbon (OC) and elemental carbon (EC)] are often a principal component of atmospheric aerosols and the atmospheric brown clouds (ABC) in South Asia (Ramanathan et al., 2005). Sources and chemical transformations of OC are complicated, including primary emissions (biomass and fossil fuel combustion, plant debris and soil dust) and secondary formation of the oxidative products of precursor gases emitted from both biogenic and anthropogenic sources (Simoneit et al., 2004;Claeys et al., 2004;Ding et al., 2016a). While past studies of aerosol characteristics in the Kathmandu Valley and surrounding regions are limited, they demonstrated that OC was the main component of aerosols observed in the Kathmandu Valley (Shakya et al., 2017;Kim et al., 2015). These studies have focused on a few aerosol species or a limited number of organic compound classes (Kim et al., 2015;Chen et al., 2015;Sarkar et al., 2016). Analysis of organic aerosols at the molecular level has been reported only at a rural Godavari site on the southern edge of the Kathmandu Valley (Stone et al., 2010;Stone et al., 2012). Overall, the composition and sources of OC are still poorly characterized.

Therefore, to overcome such research gaps, our study comprehensively investigates the organic molecular compositions of aerosols from the Kathmandu Valley, including anhydrosugars, monosaccharides, sugar alcohols, phenolic compounds, resin acid, phthalate esters, and secondary organic aerosols based on the analysis of various organic tracers emitted from primary sources and secondary formation of organic aerosol. We also studied the seasonal variation and molecular distribution to decipher their abundances, understand their predominant sources (primary vs. secondary), and to quantify the contributions of the most abundant sources of the carbonaceous aerosols in the suburban environment in the Himalayan foothills. Our work adds to the growing database on the chemical characteristics of organic aerosols in South Asia, particularly in the HKHTP region.

**2. Materials and methods**

## 2.1. Sampling site

The Kathmandu Valley is a round flat basin in the southern foothills of the Himalayas. The basin bottom has an elevation of approximately 1300 meters above sea level, and it is surrounded by a ring of green mountains, with elevation ranging from 1500 m to 2800 m above sea level (Panday and Prinn, 2009). Our sampling was performed for one year from April 2013 to April 2014 at Bode (27.67°N, 85.38°E, 1320 m above sea level), a suburban site to the east of Kathmandu city (Fig. 1). Bode is located in the Madhyapur-Thimi area in the eastern part of the Kathmandu Valley. There are two major wind flows in the Kathmandu Valley: (a) west to east, from Nagdhunga-Bhimdhunga mountain pass in the west to Nagarkot-Sanga mountain pass in the east, (b) south to north, from Bagmati River corridor to the northeast direction through the central-eastern part of the Valley. These two airflows meet around central-eastern part of the Valley and move eastward towards the Nagarkot-Sanga passes (Regmi et al., 2003;Panday and Prinn, 2009). The Bode area receives these two air flows, and hence it is downwind of Kathmandu city and Lalitpur or Patan city located in southwest, west and northwest direction during the daytime, and Bhaktapur city located in east and southeast during nighttime (Rupakheti et al., 2018;Mahata et al., 2017a;Bhardwaj et al., 2017;Sarkar et al., 2016). In addition, it is situated in a residential area with urban buildings and houses scattered across agricultural fields with paddy, wheat, corn and vegetable farms. Some small industries (plastics, electronics, wood, fabrics, etc.) and Bhaktapur Industrial Estate are located in the south-eastern direction of Bode, as well as several brick kilns that use low quality coal during January to April (Sarkar et al., 2016). The Tribhuvan international airport in the west of the Bode ($\sim$ 4 km) may have potential impacts when there is westerly wind. Approximately 1.5 and 7 km to the north, there are two reserve forests, consisting of a mix mainly broad-leaved deciduous trees and evergreen conifer trees (Department of Plant Resources, 2015). BC and $O_3$ measurements in the two major SusKat-ABC sites (Paknajol and Bode) in the Valley show similar levels (Putero et al., 2015;Mahata et al., 2017b). Therefore, Bode site can be taken as a representative site for the Kathmandu Valley (Rupakheti et al., 2018).

## 2.2 Sample collection

A medium-volume sampler (KC-120H, Laoshan Co., China) was placed on a building rooftop, about

20 m above the ground. We continuously collected total suspended particulates (TSP) for 23 h (day and night time) every five days under the flow rate of 100 L min$^{-1}$. Overall, 82 aerosol samples were successfully obtained using 90 mm diameter quartz filters (Whatman PLC, Maidstone, UK). The filters were pre-baked at 550 °C for 6 h to remove all organic material and weighed by a microbalance (sensitivity: ±0.01 mg) before and after sampling. Before each weighing, they were equilibrated for 24 h at the constant temperature (25±3 °C) and humidity (30±5%) conditions. Finally, the filters were frozen at -20 °C until laboratory analysis. Field blanks (one blank filter each month) were also collected, briefly placing a filter into the sampler without drawing air to assess potential contamination. There may be positive and negative artifacts during the sample handling/conditioning due to the adsorption/evaporation processes of organic aerosols (Fu et al., 2010;Li et al., 2018;Boreddy et al., 2017;Oanh et al., 2016). In a comparable study, Ding et al. (2013) reported the positive artifacts of 10−20% for OC and up to 16% for organic tracers using a backup quartz filter placed behind the main quartz filter.

## 2.3 Chemical Analysis

The aerosol samples were analyzed for major ions, OC, EC, and organic molecular tracers in the laboratory. Major ions ($Ca^{2+}$, $Na^+$, $K^+$, $Mg^{2+}$, $NH_4^+$, $Cl^-$, $SO_4^{2-}$ and $NO_3^-$) were measured using an ion chromatography (Dionex, USA) with ICS-320 and ICS-1500 (Tripathee et al., 2017). The detection limit for all the major ions was 0.01 µg m$^{-3}$. They denoted less than 5% of the real sample concentrations in the field blank filters (Tripathee et al., 2017). Non-sea-salt $Ca^{2+}$ (nss-$Ca^{2+}$) and $K^+$ (nss-$K^+$) was estimated according to the method from George et al. (2008). OC and EC were determined by the DRI OC/EC analyzer (Model 2001A, USA) according to the thermal/optical reflectance (TOR) method with the IMPROVE-A protocol (Wan et al., 2015). The concentrations of OC and EC from field blank filters were 0.59±0.13 µg m$^{-3}$ and 0.00 µg m$^{-3}$, respectively. The OC data reported here were blank corrected.

Detailed analytical method of organic molecular tracers was described previously by Wan et al. (2017). Briefly, small filter aliquots (1.13-3.39 cm$^2$) were cut, spiked with appropriate amounts methyl-β-D-xylanopyranoside (MXP, 99%, Sigma) and $D_3$-malic acid (DMA, CDN isotopes, 99%) as internal recovery standards. The cut filters were then extracted three times with dichloromethane/methanol (2:1, v/v) at room temperature for 30 minutes (20 ml each time). The solvent extracts in total of 60 ml were combined and successively filtered with quartz wool, concentrated, blown to dryness using ultrapure

nitrogen gas and then reacted with 50 μl of 99% N,O-bis-(trimethylsilyl)trifluoroacetamide (BSTFA, with 1% trimethylsilyl chloride) and pyridine (v/v=2:1) at 70 °C for 3 h. After derivatization, 150 μl n-hexane was added to the derivatives. A trace gas chromatography coupled to a PolarisQ mass spectrometry detector (GC-MS, Thermo Scientific) was used for analysis. The GC instrument was equipped with a TG-5MS (30 m × 0.25 mm I.D. × 0.25 μm film thickness). The injection port was set for split/splitless injection. The oven temperature was initially held at 50 °C for 2 min, increased to 120 °C at 30 °C min$^{-1}$, then to 300 °C at 6 °C min$^{-1}$ and finally held for 16 min. The MS was operated in electron ionization (EI) mode at 70 eV with a scan range of 50-650 Da.

For quantitative analysis, calibration curves were established by using authentic standards that were processed as described above. For the quantification of target compounds that were no available standards, they were estimated by the following surrogate compounds: erythritol for 2-methylglyceric acid, 2-methyltetrols and C$_5$-alkene triols, cis-pinonic acid for 3-hydroxyglutaric and 3-methyl-1,2,3-butanetricarboxylic acids, pinic acid for ß-caryophyllinic acid, azelaic acid for 2,3-dihydroxy-4-oxopentanoic acid (DHOPA). Recoveries for target compounds and MXP (Table S1) were more than 75%. The exception was for malic acid (50.3%-90.5%) and cis-pinonic acid (60.2%-81.8%). Duplicate analysis showed that the relative differences were less than 15%. The method detection limits were 0.04-0.13 ng m$^{-3}$ (Table S1). The results reported in the current study were not corrected for the recoveries. Field blank filters were analyzed by the procedure used by the samples above, but no target compounds were detected.

Table 1 shows a list of chemical species (OC, EC and major ions) and molecular markers (from the sources of biomass burning, fungal spores, plant debris, plastic emission, biogenic and anthropogenic secondary formation) analyzed in this study. They include anhydrosugars (levoglucosan, mannosan and galactosan), primary saccharides (sucrose, glucose, trehalose and xylose) and sugar alcohols (mannitol, arabitol, sorbitol and erythtitol), lignin and resin pyrolysis products (vanillic, syringic, p-hydroxybenzoic and dehydroabietic acids), phthalate esters, isoprene tracers (2-methylglyceric acid, 2-methylthreitol, 2-methylerythritol, cis-2-methyl-1,3,4-trihydroxy-1-butene, 3-methy-2,3,4-trihydroxy-1-butene and trans-2-methyl-1,3,4-trihydroxy-1-butene), monoterpene tracers (cis-pinonic, pinic, 3-hydroxyglutaric and 3-methyl-1,2,3-butanetricarboxylic acids), ß-caryophyllinic and 2,3-dihydroxy-4-oxopentanoic acids.

**2.4 Estimation of measurement uncertainty**

Since there is no commercial standard available for most SOA tracers (except for cis-pinonic acid and pinic acid), the use of surrogate standards for quantification introduces additional error to the measurements. Error in analyte measurement (EA) is propagated from the standard deviation of the field blank (EFB), error in spike recovery (ER) and the error from surrogate quantification (EQ):

$$EA = \sqrt{EFB^2 + ER^2 + EQ^2}$$

EFB was 0 in this study due to SOA tracers that were not detected in the field blanks. The spike recoveries of surrogate standards were used to estimate the ER of tracers, ranging from 9.2% (erythritol) to 26.1% (cis-pinonic acid). According to Stone et al. (2012), there is an empirical approach to estimate EQ based on homologous series of atmospherically relevant compounds. The relative error introduced by each carbon atom (En) was estimated to be 15 %, each oxygenated functional group (Ef) to be 10% and alkenes (Ed) to be 60%. Therefore, the EQ are calculated as:

$$EQ = En\Delta n + Ef\Delta f + Ed\Delta d$$

where $\Delta n$, $\Delta f$ and $\Delta d$ are the difference of carbon atom number, oxygen-containing functional group and alkene functionality between a surrogate and an analyte, respectively.

The estimated uncertainties in tracer measurement is presented in Table S2. The EQ ranged from 15% (2-methyltetrols) to 120% (β-caryophyllenic acid) in this study. Propagated with the error in recovery, EA were estimated in the range of 17.6% to 122.4%."

## 2.5 Meteorological parameters

The meteorological parameters such as air temperature (T), relative humidity (RH), atmospheric pressure (P), visibility (V), wind speed (WS) and direction (WD) were used in this study. They were derived from Tribhuvan International Airport (www.wunderground.com), which was located west of Bode (approximately 4 km). Mixing layer height (MLH) data was measured with a Vaisala ceilometer at Bode site (Mues et al., 2017). The meteorology of Kathmandu Valley and its surrounding regions is controlled by the South Asian monsoon circulations in the wet season (monsoon, June-September). Westerlies dominate the atmospheric circulation patterns during the dry seasons including pre-monsoon (March-May), post-monsoon (October-November) and winter (December-February) with limited precipitation (Pudasainee et al., 2006;Mues et al., 2017). Additionally, the valley is also influenced by

local mountain valley circulation (Mues et al., 2018).

## 3. Results and discussion

A statistical concentration summary of constituent major ions, OC, EC, and major organic compounds identified in TSP samples collected at the Bode site is presented in Table 1. Tracers for six classes of organic compounds were detected: anhydrosugars, monosaccharides, sugar alcohols, phenolic compounds and resin acid, phthalate esters, and secondary organic aerosol (SOA) tracers.

### 3.1 Aerosol loadings

The TSP samples at Bode site exhibited daily mass concentrations from 32.0 to 723 µg m$^{-3}$ with an average concentration of 255±167 µg m$^{-3}$ during April 2013 to April 2014 (Table 1). Putero et al. (2015) reported 195±83 µg m$^{-3}$ of online PM$_{2.5}$ concentration in Pakajol site (also one of SusKat-ABC sites), accounting for roughly 80% of TSP in our study. The TSP concentrations were comparable to those reported in other heavily polluted cities in South Asia, including Islamabad in Pakistan (Shah and Shaheen, 220  2008), Kolkata (Gupta et al., 2007) and Agra (Rajput and Lakhani, 2010) in India. Compared to the remote sites such as Manora Peak in the central Himalaya (Ram et al., 2010) and Lulang in the Tibetan Plateau (Wang et al., 2015), the TSP in Bode shows significantly higher mass concentrations. We found a clear seasonal variation in TSP mass concentrations (Fig. 2a) with a maximum during the pre-monsoon season while a minimum during the monsoon season, and an increasing trend from the post-monsoon to the 225  winter season were observed. It generally corresponded to the build-up of the atmospheric brown clouds (ABCs), which engulfed most of South Asia and the Northern Indian Ocean, during the long dry season extending from November to May (Ramanathan et al., 2005).

    Meteorological parameters may also affect the TSP concentrations. The highest TSP concentration observed during the pre-monsoon can be caused by the fugitive dust which is been blown up by strong 230  wind and the absence of wet-precipitation (Fig S1a and c). The lower TSP concentration in the monsoon was likely related to increased precipitation (Fig S1c) after the onset of the South Asian monsoon. Nearly 80% of the annual precipitation falls during the monsoon season in the Kathmandu Valley, which flushes out pollutants from the atmosphere (Sharma et al., 2012;Tripathee et al., 2017). During winter, an inversion layer often occurs in the Kathmandu Valley owing to its bowl-shaped topography (Pudasainee

et al., 2006). The existence of an inversion layer with the lower temperature ($12.0\pm2.41$°C), wind speed ($2.86\pm1.34$ km h$^{-1}$), and MLH ($0.34\pm0.08$ km) (Mues et al., 2017) (Fig S1a, c and d) altogether reduced the pollution dispersion mechanism resulting in increased levels of pollutants close to the ground surface.

## 3.2 Major ions and OC/EC

Concentrations of eight major ions were measured in the aerosol samples from the Bode site. The total sum accounted for $17.1\% \pm 8.5\%$ of annual average TSP mass. Sulfate ranked the highest among them, with an annual mean of $10.8\pm9.83$ µg m$^{-3}$, followed by $Ca^{2+}$ ($7.96\pm6.85$ µg m$^{-3}$), $NH_4^+$ ($5.92\pm6.16$ µg m$^{-3}$), $NO_3^-$ ($5.21\pm4.35$ µg m$^{-3}$), $Na^+$ ($3.28\pm1.58$ µg m$^{-3}$), $K^+$ ($2.43\pm2.82$ µg m$^{-3}$), $Cl^-$ ($2.15\pm2.25$ µg m$^{-3}$) and $Mg^{2+}$ ($0.61\pm0.54$ µg m$^{-3}$). On average, the combination of $SO_4^{2-}$, $NO_3^-$ and $NH_4^+$, i.e. the secondary inorganic aerosols, constituted more than half (51.3%) of the total ionic concentrations. The $Ca^{2+}$ alone accounted for 22.1% of total ions.

Sulfate, ammonium and nitrate revealed a typical seasonality with the seasonally averaged concentrations ranked in the descending order of winter > pre-monsoon > post-monsoon > monsoon. This is consistent with the seasonal variation of the precursors $NO_x$, $NO_2$ and $SO_2$, which are mainly caused by automobile exhaust, household cooking, and operation of the typical biomass co-fired brick kilns in the Kathmandu Valley (Kondo et al., 2005;Kiros et al., 2016). Currently, nearly 50% of the total motor vehicles in Nepal (approximately 2.33 million) run on the Kathmandu Valley roads (DoTM, 2015;Mahata et al., 2017b)**.** Diesel- or gasoline-powered generators (producing higher $NO_x$ emissions) and garbage burning are other major sources of air pollution in Nepal during the sampling period, which can also emit many aerosol precursors (Stockwell et al., 2016).

Ions derived from crustal sources, such as $Ca^{2+}$ and $Mg^{2+}$ are related to the local fugitive dust sources such as road dusts and construction activities (Ram et al., 2010). Interestingly, good correlations between $Ca^{2+}$ and $SO_4^{2-}$ ($R^2 = 0.48$, P<0.001), $Ca^{2+}$ and $NO_3^-$ ($R^2 = 0.58$, P<0.001), $Ca^{2+}$ and $NH_4^+$ ($R^2 = 0.62$, P<0.001), $Mg^{2+}$ and $SO_4^{2-}$ ($R^2 = 0.61$, P<0.001), $Mg^{2+}$ and $NO_3^-$ ($R^2 = 0.71$, P<0.001), $Mg^{2+}$ and $NH_4^+$ ($R^2 = 0.69$, P<0.001) were observed (Table 2), which hinted that dust may co-exist with $SO_4^{2-}$, $NO_3^-$ and $NH_4^+$ in the Kathmandu Valley (Tripathee et al., 2017).

The annual mean concentrations of carbonaceous aerosols (OC: $38.7\pm32.7$ µg m$^{-3}$ and EC: $9.92\pm5.33$

μg m$^{-3}$) accounted for 19.2%±5.48% of TSP mass at the Bode site, which was higher than that of the major ions. OC alone accounted for 14.6%±4.81% of the TSP mass. OC and EC showed much higher concentrations during winter and pre-monsoon seasons than that in monsoon season (Fig. 2b and c). In this study, we found that the daily OC/EC ratios ranged from 0.77 to 15.8, with an annual mean ratio of 3.78±2.73, and seasonal mean ratios of 4.44, 2.71, 3.31, and 5.86 during pre-monsoon, monsoon, post-monsoon, and winter, respectively (Table 1 and Fig. 2d). The OC/EC mass ratio of more than 2.0 indicates the presence of secondary organic matter or biomass burning aerosols (Cao et al., 2007). Their influence and contribution will be discussed in the following sections. The OC/EC ratios found in this study for the Kathmandu valley were similar to other sites in South Asia, like Lumbini (5.16±2.09, 2.41-10.03) (Wan et al., 2017), Delhi (5.86±0.99, 2.9-9.2) (Bisht et al., 2015) and Lahore (3.9 ± 1.6,1.5-7.2) (Alam et al., 2014).

### 3.3 Sugar compounds

### 3.3.1 Anhydrosugars

Levoglucosan (1,6-anhydro-β-D-glucopyranose) and its two isomers (mannosan and galactosan) have been used as an ideal molecular tracer for biomass burning emissions (Graham et al., 2002;Simoneit, 2002). They are exclusively produced from the pyrolysis of cellulose and hemicellulose. In the current study, levoglucosan was observed as the most abundant species among the individual compounds identified with an average concentration of 788±685 ng m$^{-3}$ (ranging from 58.8 to 3079 ng m$^{-3}$) (Table 1).

For the seasonality, levoglucosan showed significantly higher levels during winter, pre-monsoon and post-monsoon (Fig. 3a). Especially, the higher concentrations were recorded in winter ranging from 830 to 2395 ng m$^{-3}$ (average: 1391±535 ng m$^{-3}$). It showed the comparable levels with other sites in the world, which were badly affected by the biomass burning emissions, e.g., New Delhi (1977 ng m$^{-3}$) and Raipur (2180 ng m$^{-3}$) in India (Li et al., 2014;Deshmukh et al., 2016), Tasmania (4540 ± 2480 ng m$^{-3}$) in Australia (Reisen et al., 2013) and Lumbini (1161±1347) in Nepal (Wan et al., 2017). Our results were much higher than the aerosols (20-372 ng m$^{-3}$) collected at a rural Godavari site, located on the southern edge of the Kathmandu Valley during 2006 (Stone et al., 2010). Good correlations exhibited between levoglucosan and OC (R$^2$=0.79, P<0.001), EC (R$^2$=0.42, P<0.001) and nss-K$^+$ (R$^2$=0.35, P<0.01) during

the campaign (Fig. 4). This indicates that OC and EC in Kathmandu Valley's aerosols are strongly related to biomass burning source (Kim et al., 2015).

The ratio of levoglucosan to mannosan (Lev/Man) has been applied to distinguish the possible biomass burning categories. Higher Lev/Man ratios were reported for emissions from combustion of hardwood (ranging from 12.9 to 35.4 with an average of $21.5 \pm 8.3$) and agricultural residues (range from 12.7 to 55.7 with an average of $32.6 \pm 19.1$) in previous studies (Engling et al., 2009;Sang et al., 2013). For the softwood burning, the average ratio was $4.0 \pm 1.0$ (ranging from 2.5 to 5.8). In this study, the average Lev/Man ratio was 16.3±5.96 (ranging from 9.13 to 33.1 with only 9 samples <10). It can be inferred that the combustion of crop residues and hardwood is likely to be one of the major sources of atmospheric pollution in this region. A previous study also reported that the combustion of wood fuel for cooking and heating is common during wintertime in Nepal while there is much more crop residue combustion during pre-monsoon and post-monsoon seasons (Stockwell et al., 2016). This is not only a local but also a regional phenomenon; for example, Bhardwaj et al. (2017) and Wan et al. (2017) reported that the emissions from crop residue burning during the pre-monsoon and post-monsoon seasons from western India and eastern Pakistan impact the air quality in Nepal. Similarly, Rupakheti et al. (2017) also showed that the combustion of agricultural residues and forest fires over the northwestern IGP region are causes of the air pollution episodes over the foothills of the central Himalayas. In addition, brick kilns mainly operated during January-April, burned substantial quantities of low-grade coal, mixed crop wastes and firewood (Stone et al., 2010;Kim et al., 2015). Such emissions may also lead to the high levels of levoglucosan observed at Bode. We must point out that the incense burning in Kathmandu Valley may also influence the levoglucosan concentration.

### 3.3.2 Monosaccharides

Primary biological aerosol particle (PBAP) tracers, commonly known also as bioaerosols, were analyzed in the Bode aerosol samples, including five monosaccharides of glucose, fructose, trehalose, sucrose and xylose. PBAP is derived from fungal spores, vegetative debris, pollen, bacteria and viruses. Most of them can cause allergenic or immunotoxic effects on human health (Bauer et al., 2008;Myriokefalitakis et al., 2017).

In this study, total monosaccharides had an annual mean concentration of $298\pm127$ ng m$^{-3}$. Glucose was the most abundant species ($124\pm60.0$ ng m$^{-3}$), followed by fructose ($58.2\pm28.3$ ng m$^{-3}$), sucrose ($48.3\pm27.4$ ng m$^{-3}$), trehalose ($40.8\pm22.0$ ng m$^{-3}$), and xylose ($26.5\pm18.1$ ng m$^{-3}$) (Table 1). Except xylose, they all presented higher concentrations in pre-monsoon while being lower in winter (Fig 3h, i, j and k). There were significant linear correlations between glucose and fructose ($R^2 = 0.77$, $p<0.001$), trehalose and glucose ($R^2 = 0.30$, $p<0.001$), trehalose and fructose ($R^2 = 0.23$, $p<0.001$), sucrose and glucose ($R^2 = 0.55$, $p<0.001$), sucrose and fructose ($R^2 = 0.55$, $p<0.001$), and sucrose and trehalose ($R^2 = 0.28$, $p<0.001$) (Table 3). Therefore, the strong correlations indicated that they were derived from common sources, e.g. from local forests in the Kathmandu Valley during the period of high productivity of plants. In addition, the pollen produced from the flowering of local vegetation also largely contribute to glucose, fructose, trehalose, sucrose. The flowering of trees and crops peaks during the pre-monsoon season. The similar phenomenon was also reported in deciduous forests in northern Japan (Miyazaki et al., 2012).

Xylose has complex sources, including soils (Simoneit et al., 2004), microbiota (Wan and Yu, 2007), vegetation, bacteria (Cowie and Hedges, 1984) and biomass combustion (Zhu et al., 2015). It presents less abundant and only accounts for $6.90\%\pm8.32\%$ of the total PBAP tracers identified in the Bode aerosols. For the seasonal pattern, it is characterized by waxing in winter ($47.4\pm24.3$ ng m$^{-3}$) and waning in monsoon ($12.0\pm5.16$ ng m$^{-3}$), which was different from the other primary monosaccharides (Table 1 and Fig. 2l). Close correlation between xylose and levoglucosan (the biomass burning tracer) was observed in our study (Fig. S2, $R^2=0.72$, $p<0.001$), indicating that the emissions from the burning of biomass may largely contribute to xylose in Bode aerosols. A similar finding of xylose source (i.e. biomass burning) was also proposed by Zhu et al. (2015).

### 3.3.3 Sugar alcohols

Total concentration of sugar alcohols (arabitol, sorbitol, erythtitol and mannitol), was $213\pm126$ ng m$^{-3}$, and thus lower than that of total monosaccharides (Table 1). Mannitol ($86.9\pm55.3$ ng m$^{-3}$) and arabitol ($68.4\pm39.8$ ng m$^{-3}$) were the most abundant species of the sugar alcohols, followed by erythitol ($43.1\pm28.8$ ng m$^{-3}$) and sorbitol ($14.2\pm8.02$ ng m$^{-3}$). All of them exhibited monsoon maxima ($114\pm61.4$ ng m$^{-3}$, $86.6\pm44.5$ ng m$^{-3}$, $56.9\pm33.1$ ng m$^{-3}$ and $17.9\pm9.31$ ng m$^{-3}$, respectively) and winter minima ($18.1\pm6.02$

ng m$^{-3}$, 26.1±9.13 ng m$^{-3}$, 5.82±2.72 ng m$^{-3}$, 12.4±7.60 ng m$^{-3}$, respectively) (Table 1 and Fig. 3m, n, o and p). They also showed significant correlations with each other, implying their common sources (Zhu et al., 2015;Fu et al., 2012). Mannitol and arabitol have been mostly associated with fungal spores, along with vegetation and mature leaves and algae (Liang et al., 2016;Myriokefalitakis et al., 2017;Yttri et al., 2007). Recent studies proposed that elevated concentrations of mannitol and arabitol were usually observed augmentation after rain events and also highly correlated with relative humidity (Zhang et al., 2010;Yue et al., 2016). Therefore, at Bode, sugar alcohols were likely emitted by plants in nearby forest and agriculture fields, especially during the monsoon with the higher relative humidity (Fig. S1b). In addition, the higher temperatures (Fig. S1a) were conducive for more active microbial activities. Notably, the levels of PBAP discussed above were much higher than other sites in the world (Zhu et al., 2015;Chen et al., 2013;Liang et al., 2016), indicating the strong fungal spore production in the Kathmandu Valley during the monsoon season.

## 3.4 Phenolic compounds and resin acid

Phenolic compounds (e.g., vanillic, syringic and p-hydroxybenzoic acids) derived from lignin pyrolysis and resin acid (e.g., dehydroabietic acid) from burning of conifer plants can be also used as biomarkers for biomass burning. Vanillic acid is dominant both in softwood and hardwood smoke (Myers-Pigg et al., 2016;Fu et al., 2010) while syringic acid is prevalent in hardwood smoke. Herbaceous plant smoke primarily contains p-anisaldehyde and p-anisic acid (e.g., p-hydroxybenzaldehyde and p-hydroxybenzoic acid). Dehydroabietic acid is a dominant compound in the total lipid material from pine wood smoke. Therefore, four pyrolysis products of lignin and resin acid (p-hydroxybenzoic, vanillic, syringic and dehydroabietic acids) were chosen as organic tracers in this study.

p-Hydroxybenzoic acid (19.8±12.3 ng m$^{-3}$) was the predominant species, followed by dehydroabietic (13.8 ±6.19 ng m$^{-3}$), vanillic (15.3±11.3 ng m$^{-3}$) and syringic acids (17.1±13.7 ng m$^{-3}$) (Table 1). They exhibited maximum concentrations during winter and pre-monsoon, decreased in monsoon and then increased from post-monsoon, which was consistent with the seasonal variation of levoglucosan (Fig 3d, e, f and g). There were also significant correlations of lignin and resin pyrolysis products with levoglucosan (cellulose pyrolysis products) (Fig. S3a, p-hydroxybenzoic acid and levoglucosan, $R^2$=0.72, P<0.001; Fig. S3b, vanillic acid and levoglucosan, $R^2$=0.86, P<0.001; Fig. S3c, syringic acid and

levoglucosan, $R^2=0.83$, P<0.001; Fig. S3d, levoglucosan and dehydroabietic acid, $R^2=0.63$, P<0.001). Such a result also shows that there are various biomass combustion sources in the valley.

The mass ratio of syringic to vanillic acids (syr/van) has recently been used to further discriminate the vegetation types burned (Fujii et al., 2015;Myers-Pigg et al., 2016;Wan et al., 2017). Myers-Pigg et al. (2016) reported that the syr/van ratios varied from 0.1 to 2.44 for combustion of hardwood and herbaceous angiosperm, while it was 0.01–0.24 for burning softwood. With regard to the aerosol samples from Kathmandu Valley, syr/van ratio was 0.94 ± 0.18 of an annual average ranging from 0.65 to 1.31, indicating that combustion of hardwood and herbaceous plant (including crop residue) are the most likely sources of biomass burning in the Valley. This finding is in agreement with the results obtained from the ratios of Lev/Man as discussed in Sect. 3.3.1.

Besides the information revealed by anhydrosugars discussed in section 3.3.1, lignin and resin biomarkers in the Kathmandu Valley's air further confirmed that biomass burning emissions is an important contributor to organic aerosols in this region, particularly during winter and pre-monsoon periods.

**3.5 Phthalate esters**

Phthalic acid esters or phthalates are extensively used as non-reactive plasticizers in the manufacture and processing of plastic products. They can be easily released into the environment from the matrix by evaporation due to their physically rather than chemically bonded to the polymer. Their potential carcinogenic and endocrine disrupting properties can affect human reproduction (Fu et al., 2010;Li et al., 2016). Diethyl, di-n-butyl and bis-(2-ethylhexyl) phthalates (DEP, DnBP and DEHP) were analyzed in current study. The annual average concentration of phthalates was 510±230 ng m$^{-3}$ (165-1520 ng m$^{-3}$) (Table 1). They showed higher concentration in pre-monsoon (Fig. S4). Fu et al., (2010) reported similar concentrations of phthalates (the total of DEP, DnBP, dimethyl, diisobutyl and di-(2-ethylhexyl) phthalates) in an Indian urban site, with 553 ng m$^{-3}$ (295-857 ng m$^{-3}$) in summer and 303 ng m$^{-3}$ (175-598 ng m$^{-3}$) in winter. In South Asia, large quantities of municipal solid wastes containing plastic products are generally disposed of in open landfills. The open burning of plastics along with other municipal solid waste is common in Nepal, and thus can also release numerous phthalate compounds into the atmosphere.

### 3.6 SOA tracers

Emissions of volatile organic compounds from vegetation (VOCs) into the atmosphere, especially isoprene, monoterpenes and sesquiterpenes occurs in large amounts. These biogenic VOCs (B-VOCs) are crucial precursors of biogenic SOA. Globally, the emissions of B-VOCs (1150 TgC/yr) consisting of 44% isoprene and 11% monoterpenes are much higher than emissions of anthropogenic VOCs (only 110 TgC/yr) (Piccot et al., 1992;Guenther et al., 1995). It should be noted, besides biogenic emissions, combustion of biomass and fossil fuel also contribute to the isoprene, monoterpenes and sesquiterpenes (Jathar et al., 2014;Sarkar et al., 2016;Sarkar et al., 2017). The measurements of gaseous VOCs in winter (December 2012 to February 2013) air in the Kathmandu Valley during SusKat-ABC campaign also showed high levels of isoprene and it was attributed (at least during high isoprene periods) mostly to biogenic emissions (Sarkar et al., 2016;Sarkar et al., 2017). It is difficult to appropriately quantify the fractions of biogenic and anthropogenic emissions of these compounds, based on ambient measurement of these species alone without measurement of biomass burning tracers such as acetonitrile and furan. The budget of isoprene emissions (500 Tg y$^{-1}$) on a global scale is dominated by vegetation (Guenther et al., 2006). Therefore, in our study, we considered the oxidation products of isoprene, monoterpenes and sesquiterpenes as the tracers of biogenic emissions and attribute their main source as biogenic emissions. This may lead to some overestimation of their contributions to SOA formation.

### 3.6.1 Isoprene SOA tracers

Six isoprene-SOA (I-SOA) tracers were identified in the Bode aerosols: 2-methylglyceric acid (2-MGA), two diastereoisomeric 2-methyltetrols (2-methylthreitol and 2-methylerythritol, 2-MTLs) and three C5-alkene triols (cis-2-methyl-1,3,4-trihydroxy-1-butene, trans-2-methyl-1,3,4-trihydroxy-1-butene, and 3-methyl-2,3,4-trihydroxy-1-butene). Their total concentrations ranged from 38.8-444 ng m$^{-3}$ (174±86.2 ng m$^{-3}$), with the maximum (236±87.2 ng m$^{-3}$) in the monsoon (Table 1). During the post-monsoon and pre-monsoon, their concentrations were similar, and a little lower than those during the monsoon (Fig. 5d) while being the lowest during winter. Their seasonal variation was in agreement with the ambient temperature (Fig S1a), which can influence the isoprene emissions and the photochemical processes (Shen et al., 2015;Wang et al., 2008). The annual average concentration was higher than the

urban sites reported from Beijing (44.3 ng m$^{-3}$) and Kunming (108 ng m$^{-3}$) (Ding et al., 2016a), even one to two orders of magnitude higher than that from global oceans and the Arctic (Hu et al., 2013). Among I-SOA tracers, 2-MTLs were the major components (51.0%$\pm$10.5%) (Fig. 6), with an annual average of 94.4$\pm$58.9 ng m$^{-3}$ (ranging from 10.9 to 270 ng m$^{-3}$). Strong correlations were exhibited between the two isomers during all the seasons (Fig. S5a), implying that they formed through the similar pathway (Shen et al., 2015;Fu et al., 2010). The daily concentration of 2-MGA ranged from 7.10-79.0 ng m$^{-3}$ with an annual average of 34.2$\pm$14.8 ng m$^{-3}$. For C5-alkene triols, the average concentration was 45.0$\pm$29.4 ng m$^{-3}$. They positively correlated with 2-MTLs (Fig. S5b), indicating they were also the oxidation products of isoprene under low-NOx conditions.

According to the reaction chamber results from Surratt et al. (2010), the formation mechanism of 2-MGA remarkably differs from 2-MTLs. 2-MGA is produced under high-NO$_x$ condition while 2-MTLs are mainly formed under low-NO$_x$ or NO$_x$-free conditions. The formation of 2-MGA can be enhanced under lower RH condition, while it is opposite for 2-MTLs (Zhang et al., 2011). During the monsoon season, due to the conducive conditions of high temperature, high RH (>70 %) (Fig. S1b), high solar radiation and fully-grown plants, the isoprene emissions was large. In addition, NOx during this season was much lower than other seasons. Therefore, 2-MGA/2-MTLs ratios exhibited the lowest values (0.20$\pm$0.08) in the aerosol samples during this wet season (Fig. 7). In contrast, 2-MGA/2-MTLs ratios increased up to 0.95 in winter, owing to the lowest temperature and RH of the whole year (Fig. 7) and the higher NO$_x$ concentration in the Kathmandu Valley (Kondo et al., 2005;Kiros et al., 2016). NO$_x$ from anthropogenic sources (industry, transportation, biomass burning in the houses as well as in the field) and meteorological conditions with reduced mixing layer heights in winter would also favor the formation of 2-MGA and subsequently increase the 2-MGA/2-MTLs ratio.

Positive correlations were observed between 2-MGA, SO$_4^{2-}$ and NO$_3^-$ (Fig. 8). Budisulistiorini et al. (2017) investigated that the concentrations of B-SOA could significantly increase as the aerosol acidity enhances based on the laboratory simulations and field observations. The remarkable influence of I-SOA by SO$_4^{2-}$ might be explained by the concerted nucleophilic addition to the key intermediates in the gas phase (e.g., isoprene epoxydiols), which is the rate-determining step in SOA formation (Xu et al., 2015;Li et al., 2018). Li et al. (2018) reported that SO$_4^{2-}$ plays a significant part in promoting aqueous phase

oxidation of I-SOA tracers. There may be the similar effect of $NO_3^-$ on the SOA formation that needs further research. Therefore, the increase of $SO_4^{2-}$ and $NO_3^-$ could effectively facilitate the ring-opening reaction of isoprene epoxydiols and the SOA formation. Thus, the higher 2-MGA in the Kathmandu Valley may be due to the abundant $SO_4^{2-}$ and $NO_3^-$ during pre-monsoon season when most of the brick kilns (more than 100) are operational. Our finding confirmed that the anthropogenic pollutants such as $SO_2$ and $NO_x$ can be conductive to accelerating the oxidation of B-VOCs and enhancing the ambient concentrations of B-SOA.

### 3.6.2 Terpene SOA tracers

Besides isoprene tracers, we also measured four monoterpene oxidation products (M-SOA tracers), including cis-pinonic (PNA), pinic (PA), 3-hydroxyglutaric (3-HGA) and 3-methyl-1,2,3-butanetricarboxylic acids (MBTCA) (Claeys et al., 2007). They are produced through the photooxidation of monoterpenes with ozone and hydroxyl radical (Iinuma et al., 2004). The annual average concentration of the total M-SOA tracers was $59.3\pm24.6$ ng $m^{-3}$ (Table 1). The concentration of M-SOA tracers was higher than those reported in the previous studies from the urban site in Kunming (annual average: $44.1\pm38.8$ ng $m^{-3}$) (Ding et al., 2016b), three cities (Ohio, Michigan and California) in North America (summer: $30.4–60.6$ ng $m^{-3}$) (Stone et al., 2009) and a forest site in Hyytiälä, Europe (summer: $15.1–33.3$ ng $m^{-3}$) (Kourtchev et al., 2005).

For the seasonal variation, relatively high concentrations of M-SOA tracers occurred during pre-monsoon and post-monsoon seasons (Fig. 5e, f, g, h, and i). Interestingly, there is intensive biomass burning in Kathmandu Valley twice a year (forest fires and crop-residue fires during April to May, and crop-residue fires during October to November) discussed in section 3.3.1 and 3.4, which may have been associated with high concentrations of M-SOA tracers. During the fires, substantial amounts of aerosols and VOCs including isoprene and monoterpenes would generate, which can enhance the levels of B-SOA tracers (Ding et al., 2013;Yan et al., 2008;Jathar et al., 2014). Good correlations were observed between levoglucosan and the higher generation oxidation products (e.g., 3-HGA and MBTCA, $R^2=0.32$ and $R^2=0.53$ respectively) in the Bode aerosols (Fig. S6). The forests in the Kathmandu Valley consist of broad-leaved evergreen mixed forest of Schima castanopsis at the base, oak-laurel forest in the middle

(1800 to 2400 m a. s. l.) and oak forest at the top, while the conifer tree species Pinus roxiburghii (Khote Salla) and Pinus wallichiana (Gobre Salla) are also found (Department of Plant Resources, 2015;Sarkar et al., 2016). Monoterpenes were chiefly emitted from needle leaf trees (coniferous trees) (Kang et al., 2018). Therefore, it suggested that biomass-burning activities have had a significant influence on the atmospheric composition over Kathmandu Valley, especially for SOA tracers.

Sesquiterpenes are also among the biogenic SOA (B-SOA) precursors emitted from trees, which have been observed in the troposphere in a variety of field studies. Concentrations of ß-caryophyllenic acid found in the Bode aerosols ranged from 1.53 to 18.5 ng m$^{-3}$ with an average of 6.31±3.86 ng m$^{-3}$. It shared the similar seasonal variation with M-SOA tracers and positively correlated with them, indicating the possible common emission pattern.

### 3.6.3 Aromatic SOA tracer

Anthropogenic SOA is also an important OC source. 2,3-dihydroxy-4-oxopentanoic acid, DHOPA is a tracer of anthropogenic SOA from aromatics. In this study, the level of DHOPA was higher during winter and pre-monsoon while lower during monsoon season (Fig. 5). Though the major emissions of aromatics come from solvent and fossil fuel use, biomass burning is also considered as an possible source in some sites of the world (Shen et al., 2015). There was a good correlation between DHOPA and levoglucosan (Fig. 9), especially during pre-monsoon with the value of R$^2$ at 0.73. This indicated that biomass-burning emission is an important source of DHOPA at the Bode site.

### 3.7 Estimation of the contributions of different sources to OC

As discussed above, both the primary and secondary sources have influence on OC in the atmospheric aerosols of the Kathmandu Valley. In this part, we will apply the tracer-based methods to estimate the contributions of different sources to OC. It should be noted here that tracer methods can provide a reasonable estimation, but uncertainties are introduced considering the site differences and the lack of representative source profiles for the given study location. The contribution evaluated from each source to OC in the current study is still inferable.

### 3.7.1 Biomass burning-derived OC

The ratio of levoglucosan to OC (Lev/OC) detected in source samples has been used in a wide range to quantitatively estimate the contribution from biomass burning to OC (Stone et al., 2012;Wan et al., 2017;Zhang et al., 2015), although the ratios in the BB source emissions vary among different types of biomass fuels and burning conditions (Mochida et al., 2010). Andreae and Merlet (2001) reported an average of 8.14% with a range from 8.0% to 8.2 % for Lev/OC from the burning sites of biofuel, crop residues, savanna, tropical forests, and so on. Zhang et al. (2007) reported that Lev/OC ratios ranged from 5.4%-11.8% (an average of 8.27%) in the combustion aerosols from cereal straw (wheat, corn and rice). Sheesley et al. (2003) reported an average of 7.94% of levoglucosan from the combustion of biomass (including rice straw, biomass briquettes, dried cow-dung patties, etc.) indigenous to South Asia. However, the ratio obtained from the hardwood burning in fireplaces and stoves in the US was 14%, which was applied at the background sites in Europe (Fine et al., 2004). Stone et al. (2012) used the Lev/OC ratio of 12% ±0.2% during the burning of acacia wood at Godavari in the Kathmandu Valley for the CBM profile source apportionment. The mean value of Lev/OC value of biomass burning from main biomass types was 10.1%. In this study, we choose the mostly used values of 8.14% for biomass burning estimation (Graham et al., 2002;Fu et al., 2014;Ho et al., 2014;Sang et al., 2011;Zhu et al., 2016;Mkoma et al., 2013). In addition, we also calculated the uncertainties of using different ratios (see Table S3). The diagnostic ratios among molecular tracers and OC (e.g., Lev/OC) from direct emissions are critical for more precise results. It's meaningful to understand the emission characteristics for individual OC emission categories, as well as in different locations, especially in South Asia.

Figures 10 and 11 present the monthly concentration variations of BB-OC and contribution of BB-OC to OC, respectively. Our estimation exhibited that BB-OC contributed 24.9±10.4% (ranging from 6.32% to 61.5%) to OC throughout the year in Bode aerosols (Fig. 11a). This was higher than the study in Lumbini in Nepal (19.8±19.4%) (Wan et al., 2017), and nearly twice of the BB-OC contribution to OC reported in Hong Kong (6.5%–11%) (13.1 %) and the Pearl River Delta in China (Sang et al., 2011;Ho et al., 2014). Moreover, the contribution of BB-OC to OC in current study maximized in the post-monsoon (36.3±10.4%), higher than that in the pre-monsoon (28.5±10.3%) and winter (27.9±8.63%). These results indicate that biomass burning severely affect the air quality in the Kathmandu Valley, especially during the post-monsoon season. Similarly, Stone et al. (2010) reported 21±2% of OC in $PM_{2.5}$ from Godavari

rural site in the outskirts of the Kathmandu Valley during 2006, was also attributed to the primary biomass burning sources.

### 3.7.2 Plant-debris-OC and fungal spore-derived OC

Primary biological aerosol particle (PBAP) has been identified as an important source using tracers (section 3.3.2). They are likely to have a big contribution to the aerosols in Bode. In order to reveal how much they are contributing to organic aerosols, "total" plant debris was calculated based on glucose following the equation (Puxbaum and Tenze-Kunit, 2003):

Cellulose (μg) = D-Glucose (μg)×GF×(1/SY);

Plant debris=2×cellulose.

where GF (0.90) is the glucose/cellulose weight conversion factor and SY is the saccharification yield (0.717).

For OC fraction derived from fungal spores, it was estimated using mannitol levels according to the studies from Bauer et al. (2008) and Holden et al. (2011), i.e., there was 1.7 pg mannitol and 13 pg OC per spore.

As shown in Fig. 11a, fungal spore-derived OC and plant-debris-OC annually contribute to 3.15±2.86% and 1.42±1.03% of OC, respectively. The contributions were both higher in the monsoon season, with 5.85±2.50% for fungal spore-derived OC and 2.29±0.79% for plant-debris-OC to OC, respectively (Fig. 11c). During winter, the contributions were the lowest due to the inactive vegetation. There are also some similar results from the literatures. For example, Zhu et al. (2016) reported the contribution of plant debris to OC was 5.6% in nighttime and 4.6% in daytime respectively from aerosols in a mid-latitudinal forest. Szidat et al. (2006) reported the plant debris contributed to 3.2% of OC during summer in urban aerosols collected in Zurich, Switzerland. Fungal-spore-derived OC was the biggest contributor to total OC of 3.1 % (0.03 %–19.8 %) in marine aerosols collected over the East China Sea during 18 May to 12 June 2014 (Kang et al., 2018). The study in the aerosols of Brazil urban site showed the mean contributions of fungal aerosol to OC was 8% (Emygdio et al., 2018). Liang et al. (2017) reported the contributions of fungal spores to OC of 1.2 ± 0.7% and 3.5 ± 3.7% in aerosols from an urban site and a rural site respectively during an entire year in Beijing, China. All above strengthened the

importance of plant-debris and fungal spores to the aerosol burden in the atmosphere.

### 3.7.3 Biogenic SOC and anthropogenic toluene SOC

Biogenic secondary organic carbon (B-SOC) and anthropogenic aromatic SOC (A-SOC) from the oxidation of isoprene, monoterpenes, sesquiterpene and toluene were assessed using the tracer-based method proposed by Kleindienst et al. (2007). This method has been applied successfully in many aerosol studies (Fu et al., 2010;Shen et al., 2015;Ding et al., 2016a). The mass fraction of tracer compounds in SOC (FSOC) for an individual precursor was calculate based on the smog chamber simulations. The calculation formula as following:

$$FSOC = \frac{\sum_i [\text{tri}]}{[SOC]}$$

where [tri] is the concentration of tracer i and [SOC] is the concentration of SOC. The conversion factors of FSOC were 0.155±0.039, 0.231±0.111, 0.0230±0.0046 and 0.0079±0.0026 µg µgC$^{-1}$ for isoprene, monoterpenes, sesquiterpene and toluene, respectively (Kleindienst et al., 2007).

The total calculated concentrations of B-SOC ranged from 0.41 to 2.77 µg m$^{-3}$ with an annual average concentration of 1.36±0.49 µg m$^{-3}$, a higher concentration of 1.43±0.48 µg m$^{-3}$ in monsoon and lower concentration of 0.86±0.20 µg m$^{-3}$ in winter (Fig 10g). The B-SOC/OC showed a higher average percentage of 10.1%±3.34% in the monsoon season (Fig 11c), suggesting that B-SOC was an important sources to OC at the Bode during this period. During post-monsoon, B-SOC/OC declined to 5.36% (Fig 11d). The B-SOC/OC showed the lowest value of 1.52%±0.70% in winter (Fig 11e), indicating that B-SOC had minor contributions to elevated OC in winter. The annual average concentration of A-SOC was 2.45±1.45 µg m$^{-3}$, which is higher than the B-SOC. The highest A-SOC concentration was obtained in winter (3.27±1.25 µg m$^{-3}$) (Fig.10h). A-SOC was the second most important contributor to OC after BB-OC. It is not only derived from increased fossil fuel combustion and the subsequent oxidation, but also from biomass burning emissions.

In total, SOC (including B-SOC and A-SOC) reconstructed using the formula above in this section was 3.81±1.63 µg m$^{-3}$, accounting for 15.0%±8.99% of OC.

### 3.7.4 Possible sources of the unidentified OC

On the whole, biomass burning contributed one-fourth (24.9% $\pm$ 10.4%) of the OC in Bode, followed by A-SOC (8.82% $\pm$ 5.55%), B-SOC (6.19% $\pm$ 4.49%), fungal-spores (3.15%$\pm$2.86%) and plant-debris (1.42% $\pm$ 1.03%) (Fig. 11a). Nevertheless, there is still part of OC (55.5%) that we were not able to be attributed to any specific sources based on the tracers analyzed in current study. There are partly uncertainties caused by the organic tracer analyses (estimation of measurement uncertainty was shown in Table S2). Furthermore, fossil fuel combustion and soil dust could be also notable fractions of OC in Bode aerosols. Additionally, low molecular weight (LMW) dicarboxylic acids from both primary and secondary sources also constitute a significant fraction of atmospheric organic aerosols (Kawamura and Bikkina, 2016). Humic-like substances and amines are another source of OC, but not well studied (Wu et al., 2018;Laskin et al., 2015). Therefore, the possible contributions of the unidentified OC (55.5%) from various sectors need further investigation, which is better to comprehensively understand the sources of South Asian aerosols and will be very useful for the targeted pollution control measures in this region.

### 4. Summary and conclusions

Field measurements of primary and secondary organic compounds in aerosols were conducted in Bode, a semi-urban site of the Kathmandu Valley, Nepal, from April 2013 to April 2014. A distinctive seasonality was observed for various aerosol species. Higher concentrations of OC, EC, anhydrosugars, phenolic compounds and resin acid were observed during the winter and pre-monsoon seasons, while their concentrations were lower during the monsoon season. Levoglucosan was the most abundant species among the individually identified tracers with an average concentration of 788 ng m$^{-3}$. We observed the highest abundances of monosaccharides during the pre-monsoon season and of sugar alcohols during the monsoon season, and lower levels in winter because of the reduced plant activities. I-SOA tracers represented a majority among B-SOA tracers with a maximum in the monsoon. The seasonal variation of M-SOA tracers was controlled by monoterpenes emission and biomass burning. DHOPA exhibited higher concentrations during the winter and pre-monsoon season.

The likely OC sources were further evaluated for their contributions to observed total OC using tracer-based methods. Biomass burning contributed a major fraction (24.9%) to OC in Bode, followed by

A-SOC (8.8%), B-SOC (6.2%), fungal spores (3.2%) and plant debris (1.4%). The highest contribution of BB-OC, 36.3%, occurred during post-monsoon season. A-SOC, B-SOC, fungal spores and plant debris all made larger contributions during the monsoon. The higher BB-OC and the A-SOC contributions imply that some biomass burning and anthropogenic components are widespread in the Kathmandu Valley and thus represent the main contributors affecting the regional air quality in the Kathmandu Valley region.

The present study clearly shows that the chemical constituents and sources of OC strongly vary with seasons, as a result of diverse air pollution sources in the valley across four seasons. The heavy biomass burning and the subsequent oxidant emissions are anticipated to cause larger contributions of B-SOC to OC. Understanding OC's climate impacts is a frontier area of research, because there is still a large uncertainty in the estimation of OC radiative forcing. Our study implies that since biomass burning is a major source of ambient OC, the fraction of OC that absorbs light (referred to as brown carbon) and also acts as cloud condensation nuclei, needs to be further studied in order to better understand radiative effects of OC on regional climate change. The current source contribution estimates from the tracer-based methods do not accurately evaluate the large temporal variations from all kinds of sources. Contributions from other sectors (ca 55.5%), including low molecular weight dicarboxylic acids (Kawamura and Sakaguchi, 1999;Kawamura and Bikkina, 2016), need further investigation to better understand the atmospheric aerosols from both urban and rural sources such as the Kathmandu Valley and other sites in the Himalayan foothills and the Indo-Gangetic Plain regions. These observations of the severe air pollution, particularly the particular matter pollution, provide valuable support for air pollution control measures, especially in determining which sources and sectors to first focus on the Kathmandu Valley and the surrounding region, in order to reduce the air pollution from being severe to become much cleaner in the near future. In addition, the current study based on the molecular level-source apportionment of OC in heavy polluted region from South Asia provides a much more specific quantification of source estimation for OC, which is different from previous studies based on the bulk carbonaceous aerosol using radiocarbon ($^{14}$C) measurements, PMF and CBM.

*Data availability*. Raw data are archived at the Institute of Tibetan Plateau Research, Chinese Academy

of Sciences, and are available on request by contacting the corresponding author.

*Competing interests.* The authors declare that they have no conflict of interest.

**Acknowledgments**

The authors gratefully appreciate Shyam Kumar Newar and Bhogendra Kathayat for their assistance in the sample collections, the staffs at Bode site in the Kathmandu Valley, and all individuals and groups who participated in the SusKat-ABC field campaigns. We would like to thank senior scientist Karl Epson Yttri from Norwegian Institute for Air Research for the helpful suggestions to reply the comments from the referees. This study was supported by the Strategic Priority Research Program of Chinese Academy of Sciences, Pan-Third Pole Environment Study for a Green Silk Road (Pan-TPE) (XDA20040501), the National Natural Science Foundation of China (41522103, 41421061 and 41630754) and China Postdoctoral Science Foundation (2018M630210). The coauthors from the IASS gratefully acknowledge funding from the federal ministry of education and research (BMBF) and the Brandenburg state ministry of science, research and culture (MWFK).

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

Table 1 Concentrations of TSP, major ions, OC, EC (µg m$^{-3}$) and molecular tracers in

the aerosols from Bode, Kathmandu Valley (ng m$^{-3}$).

| Compounds | Annual | | | Pre-monsoon | | | Monsoon | | | Post-monsoon | | | Winter | | |
|---|---|---|---|---|---|---|---|---|---|---|---|---|---|---|---|
| | Mean | Median | SD | Mean | Median | SD | Mean | Median | SD | Mean | Median | SD | Mean | Median | SD |
| TSP | 256 | 213 | 166 | 381 | 366 | 171 | 120 | 107 | 52.4 | 225 | 254 | 71.6 | 353 | 348 | 68.5 |
| $SO_4^{2-}$ | 10.8 | 6.15 | 9.83 | 17.2 | 16 | 7.49 | 4.1 | 2.6 | 4.04 | 4.81 | 4.24 | 2.38 | 27.3 | 24.1 | 6.79 |
| $NO_3^-$ | 5.21 | 3.8 | 4.35 | 8.82 | 8.41 | 4.41 | 2.34 | 1.85 | 1.65 | 3.52 | 3.65 | 0.92 | 9.47 | 8.11 | 4.43 |
| $NH_4^+$ | 5.92 | 3.46 | 6.16 | 8.57 | 6.71 | 5.65 | 1.99 | 1.31 | 2.58 | 3.72 | 3.65 | 1.48 | 17.5 | 15.6 | 3.06 |
| $Ca^{2+}$ | 7.96 | 5.82 | 6.85 | 11.6 | 8.98 | 8.97 | 4.47 | 3.59 | 3.66 | 6.21 | 6.01 | 1.51 | 13.8 | 15.6 | 5.02 |
| $K^+$ | 2.43 | 1.5 | 2.82 | 3.87 | 2.42 | 4.27 | 1.15 | 0.94 | 0.96 | 1.97 | 2.03 | 0.46 | 4.03 | 3.39 | 1.45 |
| $Cl^-$ | 2.15 | 1.18 | 2.25 | 2.72 | 2.27 | 1.83 | 0.73 | 0.67 | 0.32 | 1.68 | 1.78 | 0.4 | 6.94 | 7.87 | 2.23 |
| $Na^+$ | 3.28 | 2.93 | 1.58 | 3.15 | 1.68 | 2.38 | 3.3 | 3.61 | 1.18 | 2.8 | 2.84 | 0.12 | 4.21 | 4.28 | 0.49 |
| $Mg^{2+}$ | 0.61 | 0.39 | 0.54 | 0.95 | 0.67 | 0.7 | 0.32 | 0.24 | 0.25 | 0.41 | 0.4 | 0.07 | 1.07 | 1.09 | 0.39 |
| OC | 38.7 | 24.2 | 32.7 | 59.4 | 46.9 | 37.9 | 14.6 | 14.7 | 3.76 | 31.8 | 32.9 | 12.7 | 62.8 | 53.6 | 20.6 |
| EC | 9.92 | 9.34 | 5.33 | 14.4 | 13.85 | 5.24 | 5.61 | 4.76 | 1.8 | 9.37 | 9.95 | 2.19 | 11 | 9.89 | 3.48 |
| OC/EC | 3.78 | 3.09 | 2.37 | 4.44 | 3.29 | 3.23 | 2.71 | 2.54 | 0.69 | 3.31 | 3.01 | 0.93 | 5.86 | 5.3 | 1.75 |
| **Anhydrosugars** | | | | | | | | | | | | | | | |
| Levoglucosan | 788 | 631 | 685 | 1214 | 900 | 705 | 204 | 188 | 60.1 | 863 | 836 | 252 | 1391 | 1120 | 535 |
| Galactosan | 44.4 | 33.8 | 40.3 | 68.5 | 51.4 | 42.7 | 13.2 | 13.1 | 5.55 | 34.8 | 33.8 | 10.9 | 85.6 | 72.4 | 32.8 |
| Mannosan | 50.6 | 34.5 | 45.1 | 71.8 | 61.8 | 42.3 | 16 | 16.7 | 5.5 | 39.5 | 39.6 | 12.5 | 116 | 106 | 45 |
| Subtotal | 883 | 688 | 765 | 1354 | 974 | 781 | 233 | 218 | 68.2 | 937 | 922 | 273 | 1592 | 1268 | 611 |
| **Monosaccharides** | | | | | | | | | | | | | | | |
| Glucose | 124 | 114 | 60 | 137 | 118 | 65.1 | 129 | 122 | 47 | 143 | 149 | 48.7 | 39.9 | 38.7 | 12.5 |
| Fructose | 58.2 | 53.2 | 28.3 | 68.4 | 66.9 | 29.1 | 57.1 | 48.2 | 26.8 | 58.2 | 54.7 | 21 | 27 | 22.5 | 10.2 |
| Trehalose | 40.8 | 35.6 | 22 | 48 | 53.1 | 22.4 | 40.2 | 35.5 | 22.8 | 38.9 | 40.1 | 12.5 | 20.1 | 16.8 | 9.17 |
| Sucrose | 48.3 | 40.3 | 27.4 | 64.7 | 55.5 | 31.6 | 38.7 | 38.7 | 11.9 | 56.6 | 45 | 27.1 | 18.4 | 16.6 | 6.37 |
| Xylose | 26.5 | 20.5 | 18.1 | 37.8 | 30.9 | 20 | 13.2 | 13.4 | 5.68 | 24.9 | 26.5 | 9.33 | 38.6 | 37.5 | 14.1 |
| Subtotal | 298 | 285 | 127 | 356 | 333 | 141 | 278 | 249 | 93.5 | 322 | 318 | 104 | 144 | 135 | 35.8 |
| **Sugar alcohols** | | | | | | | | | | | | | | | |
| Mannitol | 86.9 | 77 | 55.3 | 84.6 | 78.8 | 38.7 | 114 | 102 | 61.4 | 63.4 | 53.6 | 34.6 | 18.1 | 19.3 | 6.02 |
| Arabitol | 68.4 | 60.6 | 39.8 | 68.9 | 65.3 | 30.3 | 86.6 | 68.9 | 44.5 | 42.6 | 33.5 | 22.5 | 26.1 | 24.7 | 9.13 |
| Sorbitol | 14.2 | 12.7 | 8.02 | 13.1 | 12.6 | 5.96 | 17.9 | 15.9 | 9.31 | 13.1 | 13.2 | 4.65 | 5.82 | 4.97 | 2.72 |
| Erythtitol | 43.1 | 36.5 | 28.8 | 35.8 | 34 | 15.4 | 56.9 | 47.5 | 33.1 | 48.6 | 39.1 | 31.8 | 12.4 | 10.2 | 7.6 |
| Subtotal | 213 | 192 | 126 | 202 | 198 | 84.8 | 275 | 245 | 143 | 168 | 151 | 80.1 | 62.5 | 61.8 | 19.7 |
| Total sugars | 1394 | 1206 | 813 | 1913 | 1570 | 919 | 787 | 727 | 249 | 1427 | 1367 | 327 | 1798 | 1462 | 651 |
| **Phenolic compounds and resin acid** | | | | | | | | | | | | | | | |
| Vanillic acid | 15.3 | 11.3 | 11.3 | 20.8 | 15.8 | 12.9 | 7.1 | 6.84 | 1.94 | 14.3 | 12.3 | 4.95 | 26.9 | 30.5 | 9.37 |
| Syringic acid | 17.1 | 11.6 | 13.7 | 23.7 | 17.1 | 15.6 | 7.82 | 7.52 | 2.67 | 13.7 | 12.5 | 4.54 | 32 | 37.8 | 12.3 |
| p-Hydroxybenzoic acid | 19.8 | 15.5 | 12.3 | 26.1 | 19.3 | 16.7 | 14.5 | 13 | 4.57 | 14.2 | 14.6 | 3.76 | 23.6 | 23.1 | 9.15 |
| Dehydroabietic acid | 13.8 | 12.6 | 6.19 | 16.3 | 15.4 | 6.52 | 10.3 | 9.7 | 2.32 | 10.9 | 9.3 | 3.1 | 21.4 | 20.1 | 6.76 |
| Subtotal | 66.1 | 49.7 | 41.4 | 86.9 | 67 | 49.2 | 39.7 | 39.6 | 9.22 | 53 | 47.5 | 15.6 | 104 | 112.8 | 36.2 |

| Compounds | Annual | | | Pre-monsoon | | | Monsoon | | | Post-monsoon | | | Winter | | |
|---|---|---|---|---|---|---|---|---|---|---|---|---|---|---|---|
| | Mean | Median | SD | Mean | Median | SD | Mean | Median | SD | Mean | Median | SD | Mean | Median | SD |
| **Phthalate esters** | | | | | | | | | | | | | | | |
| Diethyl (DEP) | 16.6 | 15.6 | 8.41 | 19.4 | 17.5 | 11.6 | 15.4 | 15.5 | 4.42 | 14.9 | 13 | 6.25 | 12.9 | 9.3 | 6.49 |
| Di-n-butyl (DnBP) | 56.2 | 48.5 | 25.6 | 63.6 | 59.8 | 30.7 | 52.2 | 48.4 | 20.9 | 55.4 | 45.3 | 22.8 | 46.4 | 41.3 | 21.8 |
| Bis-(2-ethylhexyl) (DEHP) | 438 | 378 | 200 | 495 | 466 | 239 | 407 | 377 | 162 | 431 | 353 | 177 | 361 | 322 | 170 |
| Subtotal | 510 | 444 | 230 | 578 | 545 | 276 | 474 | 445 | 184 | 501 | 408 | 204 | 420 | 372 | 196 |
| **Isoprene tracers** | | | | | | | | | | | | | | | |
| 2-Methylglyceric acid | 34.2 | 30.2 | 14.8 | 45.9 | 48.8 | 15.2 | 25.1 | 24.5 | 6.91 | 36.2 | 34.9 | 9.86 | 25 | 24.7 | 8.56 |
| 2-Methylthreitol | 30.4 | 27.3 | 19.5 | 22 | 22.8 | 10 | 45 | 44.9 | 20.6 | 27.4 | 26.3 | 7.39 | 8.4 | 7.9 | 3.83 |
| 2-Methylerythritol | 64.1 | 58.3 | 39.6 | 45 | 44.5 | 20.4 | 97.3 | 98.9 | 38 | 53 | 52 | 9.78 | 18.5 | 18.7 | 4.63 |
| 2-Methylterols[a] | 94.4 | 84.3 | 58.9 | 67 | 68.1 | 30.4 | 142 | 142 | 58.3 | 80.5 | 78.3 | 17 | 27 | 26.6 | 8.39 |
| C5-Alkene triols[b] | 45 | 39.2 | 29.4 | 30.4 | 28.8 | 13.4 | 68.8 | 69.1 | 30 | 35.1 | 30.6 | 18 | 17.5 | 15.3 | 4.88 |
| Subtotal | 174 | 160 | 86.2 | 144 | 150 | 53.4 | 236 | 220 | 87.2 | 152 | 145 | 39.3 | 69.5 | 72.4 | 19.3 |
| **Monoterpene tracers** | | | | | | | | | | | | | | | |
| cis-Pinonic acid | 26 | 24.5 | 11.6 | 32.3 | 30 | 13.8 | 21 | 19.1 | 8.04 | 28 | 29.1 | 5.92 | 20.6 | 20 | 8.01 |
| Pinic acid | 11.9 | 10.8 | 4.48 | 11.4 | 10.4 | 3.78 | 12.7 | 12.3 | 5.07 | 13.7 | 15.6 | 5.25 | 8.83 | 8.67 | 0.92 |
| 3-Hydroxyglutaric acid | 10.6 | 9.16 | 6.85 | 13.9 | 12.1 | 7.75 | 5.75 | 4.88 | 3.01 | 16 | 15.4 | 4.18 | 11.5 | 10 | 4.72 |
| 3-MBTCA[c] | 10.8 | 9.53 | 7.36 | 16.9 | 15.1 | 7.32 | 5.14 | 4.3 | 3.12 | 11.9 | 9.61 | 3.63 | 9.3 | 8.89 | 3.75 |
| Subtotal | 59.3 | 55.6 | 24.6 | 74.6 | 72.1 | 28.3 | 44.6 | 42.6 | 14.4 | 69.5 | 65.7 | 12.6 | 50.2 | 49.6 | 10.8 |
| **Sesquiterpene tracer** | | | | | | | | | | | | | | | |
| β-Caryophyllenic acid | 6.43 | 5.35 | 3.93 | 8.61 | 7.15 | 4.29 | 3.66 | 2.85 | 2 | 8.35 | 7.2 | 3.02 | 7.16 | 6.6 | 2.76 |
| Total B-SOA tracers[d] | 136 | 234 | 50.4 | 133 | 229 | 43.9 | 156 | 275 | 53.9 | 131 | 217 | 26.2 | 75.6 | 129 | 14.2 |
| **Toluene tracer** | | | | | | | | | | | | | | | |
| DHOPA[e] | 19.4 | 16.5 | 11.5 | 22.8 | 19 | 15.5 | 15 | 13.4 | 5.69 | 17.3 | 18.2 | 5.94 | 25.8 | 23 | 9.84 |
| Total SOA tracers[f] | 259 | 249 | 94.7 | 250 | 242 | 84.9 | 299 | 286 | 101 | 247 | 243 | 48.2 | 153 | 162 | 29.4 |

[a] Sum of 2-methylthreitol and 2-methylerythritol.
[b] C5-Alkene triols: 3-Methy-2,3,4-trihydroxy-1-butene, cis-2-Methyl-1,3,4-trihydroxy-1-butene and trans-2-
Methyl-1,3,4-trihydroxy-1-butene.
[c] 3-MBTCA: 3-methyl-1,2,3-butanetricarboxylic acid.
[d] Sum of 2-methylglyceric acid, 2-methylterols, C5-Alkene triols, cis-pinonic acid, pinic acid, 3-hydroxyglutaric
acid and 3-MBTCA.
[e] DHOPA:2,3-dihydroxy-4-oxopentanoic acid.
[f] Sum of 2-methylglyceric acid, 2-methylterols, C5-Alkene triols, cis-pinonic acid, pinic acid, 3-hydroxyglutaric
acid and 3-MBTC and 2,3-dihydroxy-4-oxopentanoic acid.

Table 2 Linear correlation coefficients ($R^2$) among major ions and OC, EC in aerosols in Bode, Kathmandu Valley

| | $SO_4^{2-}$ | $NO_3^-$ | $NH_4^+$ | $Ca^{2+}$ | nss-$Ca^{2+}$ | $Mg^{2+}$ | nss-$Mg^{2+}$ | $K^+$ | nss-$K^+$ | $Cl^-$ | $Na^+$ | OC | EC |
|---|---|---|---|---|---|---|---|---|---|---|---|---|---|
| $SO_4^{2-}$ | 1.00 | | | | | | | | | | | | |
| $NO_3^-$ | 0.78** | 1.00 | | | | | | | | | | | |
| $NH_4^+$ | 0.87** | 0.69** | 1.00 | | | | | | | | | | |
| $Ca^{2+}$ | 0.48** | 0.58** | 0.62** | 1.00 | | | | | | | | | |
| nss-$Ca^{2+}$ | 0.48** | 0.58** | 0.62** | 1.00** | 1.00 | | | | | | | | |
| $Mg^{2+}$ | 0.61** | 0.71** | 0.69** | 0.91** | 0.91** | 1.00 | | | | | | | |
| nss-$Mg^{2+}$ | 0.65** | 0.76** | 0.70** | 0.88** | 0.88** | 1.00** | 1.00 | | | | | | |
| $K^+$ | 0.40** | 0.59** | 0.55** | 0.78** | 0.78** | 0.85** | 0.83** | 1.00 | | | | | |
| nss-$K^+$ | 0.40** | 0.60** | 0.55** | 0.77** | 0.77** | 0.85** | 0.83** | 1.00** | 1.00 | | | | |
| $Cl^-$ | 0.67** | 0.58** | 0.76** | 0.41** | 0.40** | 0.48** | 0.50** | 0.37** | 0.37** | 1.00 | | | |
| $Na^+$ | 0.05* | 0.08* | 0.23** | 0.48** | 0.48** | 0.37** | 0.28** | 0.40** | 0.39** | 0.09* | 1.00 | | |
| OC | 0.59** | 0.61** | 0.57** | 0.32** | 0.32** | 0.48** | 0.53** | 0.36** | 0.36** | 0.48** | 0.01 | 1.00 | |
| EC | 0.35** | 0.43** | 0.24** | 0.12** | 0.12** | 0.23** | 0.27** | 0.16** | 0.16** | 0.24** | 0.02 | 0.36** | 1.00 |

*: $P<0.1$.

**: $P<0.001$.

Table 3 Linear correlation coefficients ($R^2$) among monosaccharides and sugar alcohols in aerosols from Bode, Kathmandu Valley

| | Glucose | Fructose | Sucrose | Trehalose | Xylose | Mannitol | Arabitol | Sorbitol | Erythtitol |
|---|---|---|---|---|---|---|---|---|---|
| Glucose | 1 | | | | | | | | |
| Fructose | 0.77*** | 1 | | | | | | | |
| Sucrose | 0.55*** | 0.55*** | 1 | | | | | | |
| Trehalose | 0.30*** | 0.23*** | 0.28*** | 1 | | | | | |
| Xylose | 0.001 | 0.06* | 0.19*** | 0.05* | 1 | | | | |
| Mannitol | 0.51*** | 0.51*** | 0.23*** | 0.14*** | 0.002[a] | 1 | | | |
| Arabitol | 0.44*** | 0.50*** | 0.22*** | 0.16*** | 0.0005 | 0.77*** | 1 | | |
| Sorbitol | 0.51*** | 0.53*** | 0.20*** | 0.13** | 0.0002 | 0.83*** | 0.68*** | 1 | |
| Erythtitol | 0.46*** | 0.40*** | 0.16*** | 0.13** | 0.007 | 0.77*** | 0.62*** | 0.77*** | 1 |

a Negative values indicate negative correlations.
*$p < 0.1$
**$p < 0.01$.
***$p < 0.001$.

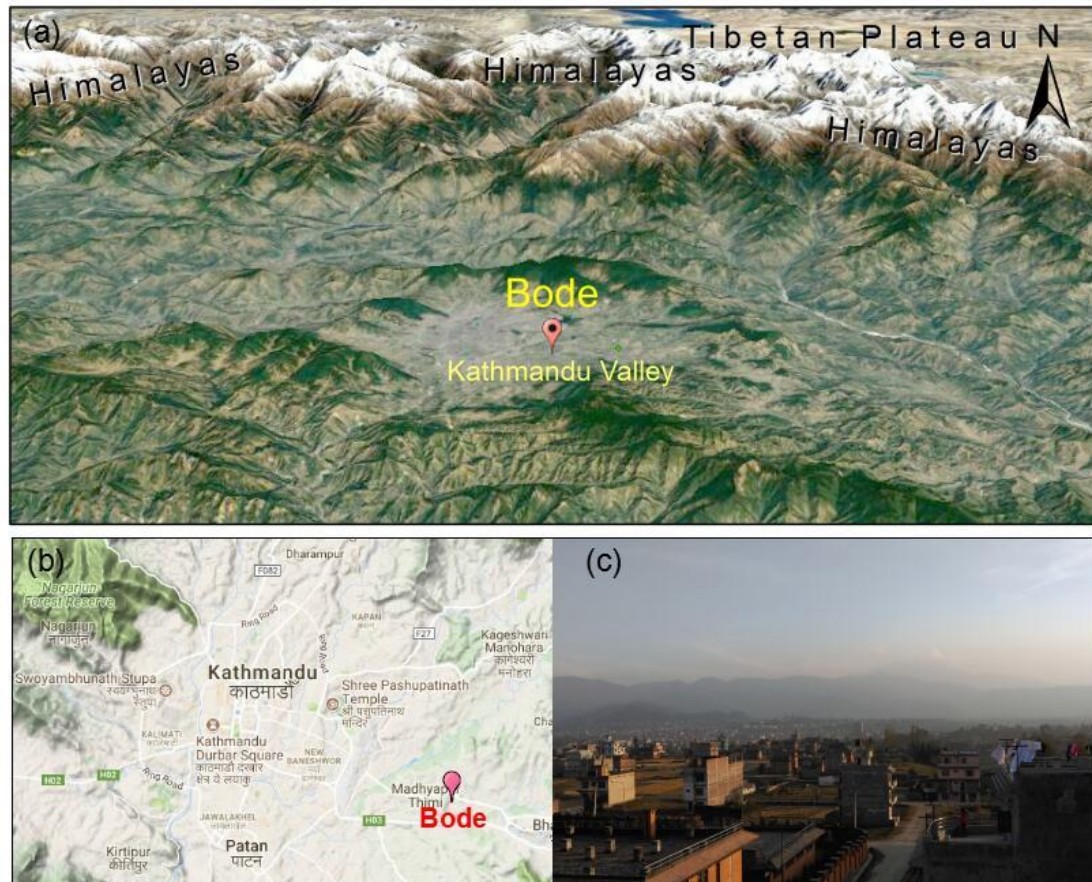

**Fig. 1.** Location of measurement site: (a) Kathmandu Valley, (b) urban measurement site at Bode in

Kathmandu Valley, (c) air pollution observed from the Bode site in the afternoon.

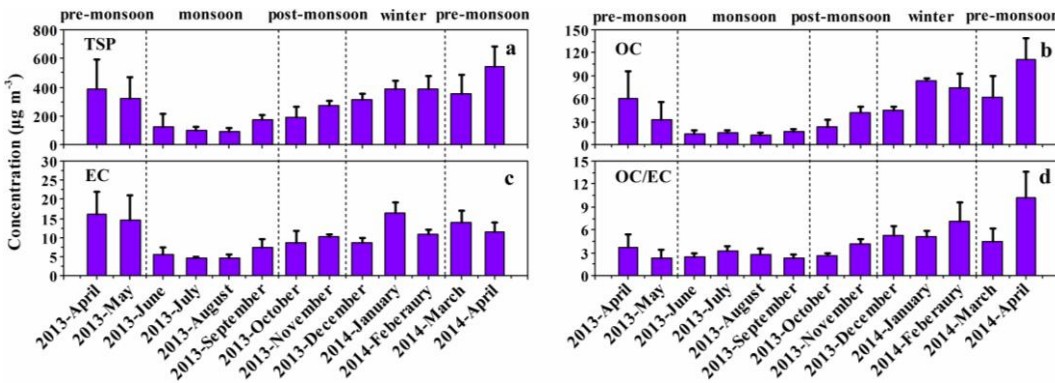

**Fig. 2.** Monthly variations of TSP, OC, EC, OC/EC ratios at Bode site, Kathmandu Valley during

April 2013-April 2014.

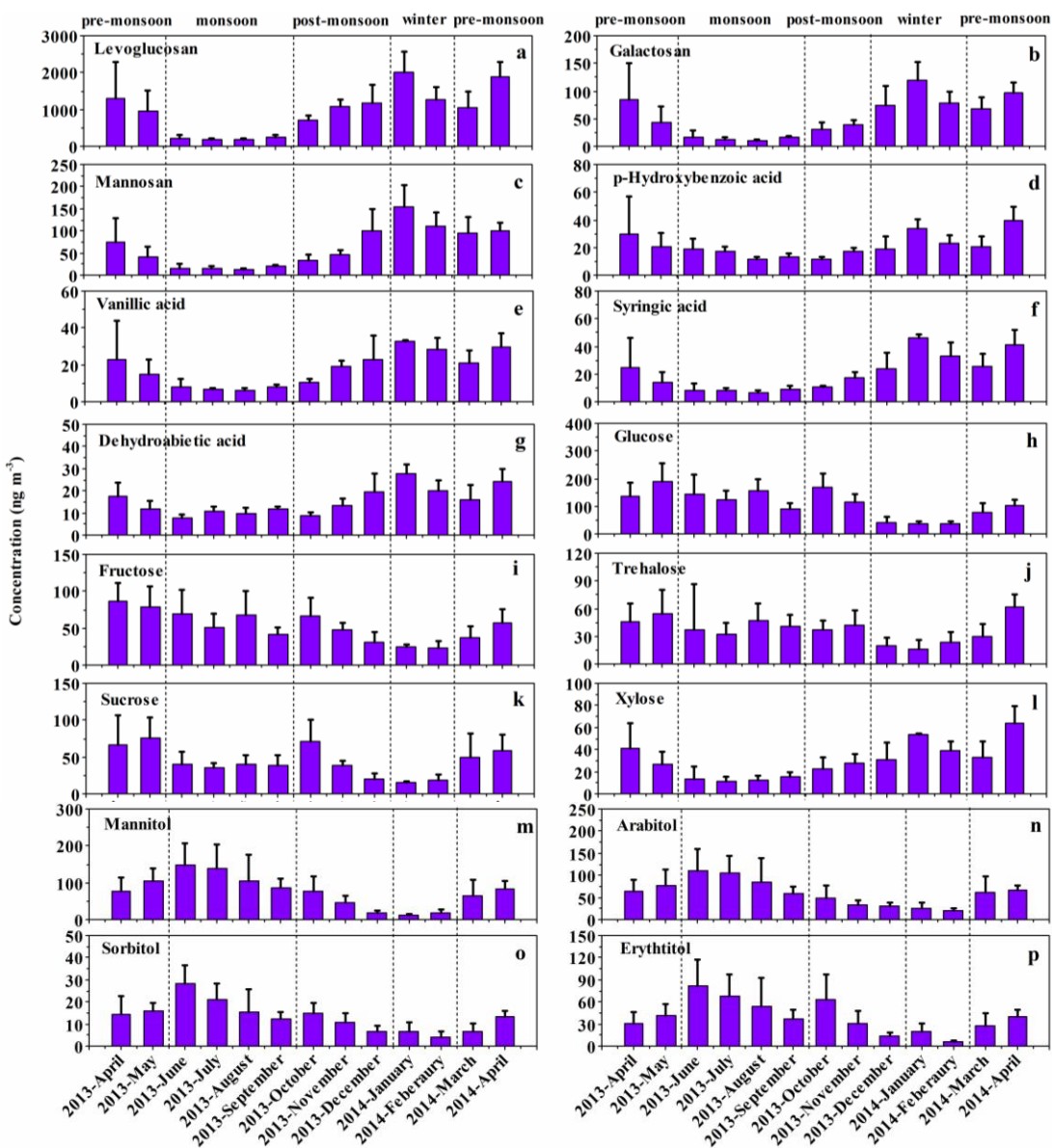

**Fig. 3.** Monthly variations of biomass burning tracers, monosaccharides and sugar alcohols at Bode site, Kathmandu Valley during April 2013-April 2014.

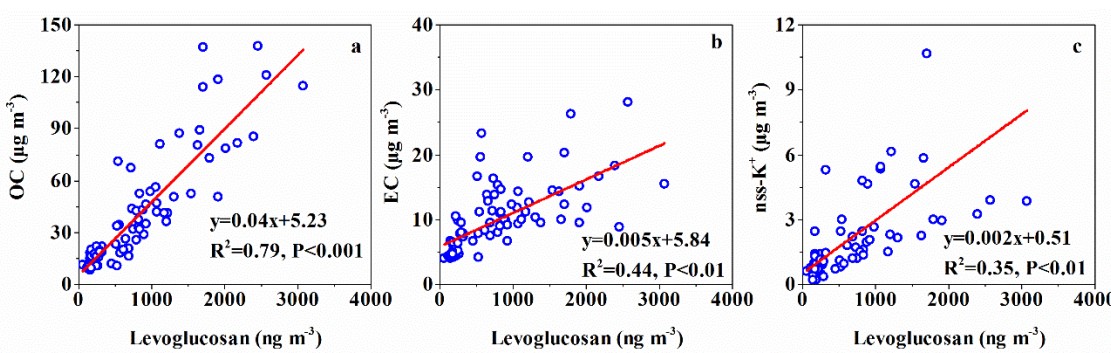

**Fig. 4.** Correlations between (a) levoglucosan and OC, (b) levoglucosan and EC, (c) levoglucosan and

nss-K$^+$ in Bode aerosols during the sampling period (April 2013 to April 2014).

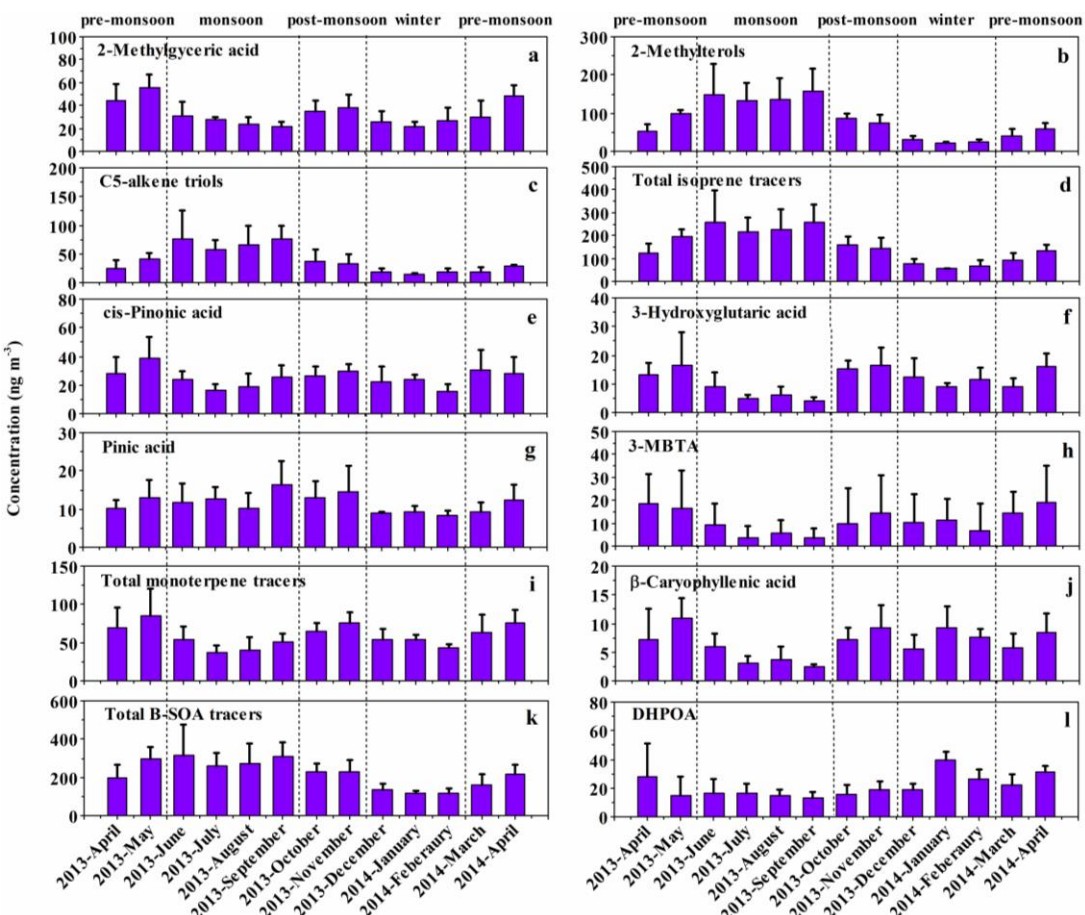

Fig. 5. Monthly variations of B-SOA tracers, total isoprene tracers, total monoterpene tracers, ß-caryophyllenic acid, total B-SOA tracers and DHPOA at Bode site, Kathmandu Valley during April 2013-April 2014.

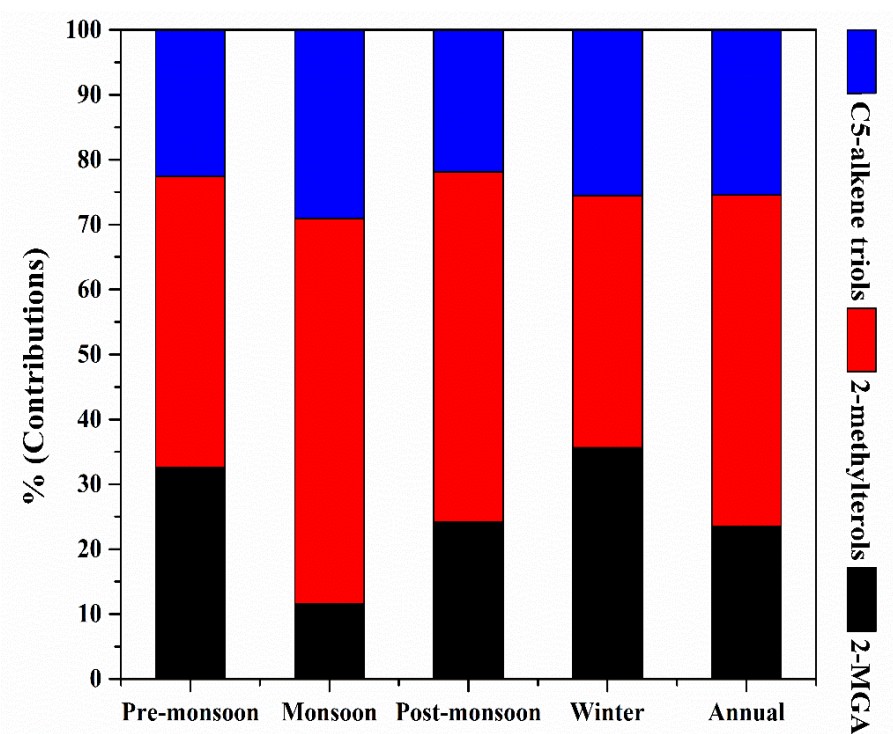

**Fig. 6.** The percentage contributions of the isoprene SOA tracers to the total during different seasons in
the atmospheric aerosols from Kathmandu.

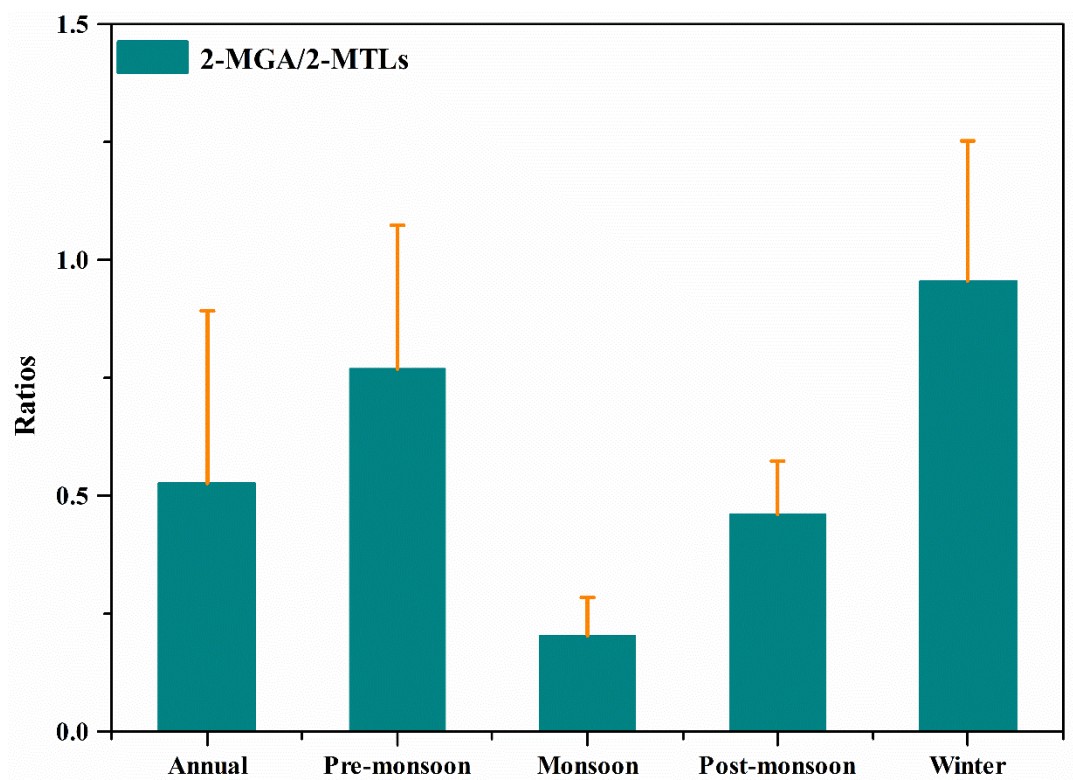

**Fig. 7.** Ratios of 2-MGA/2-MTLs during different seasons in Bode, Kathmandu.

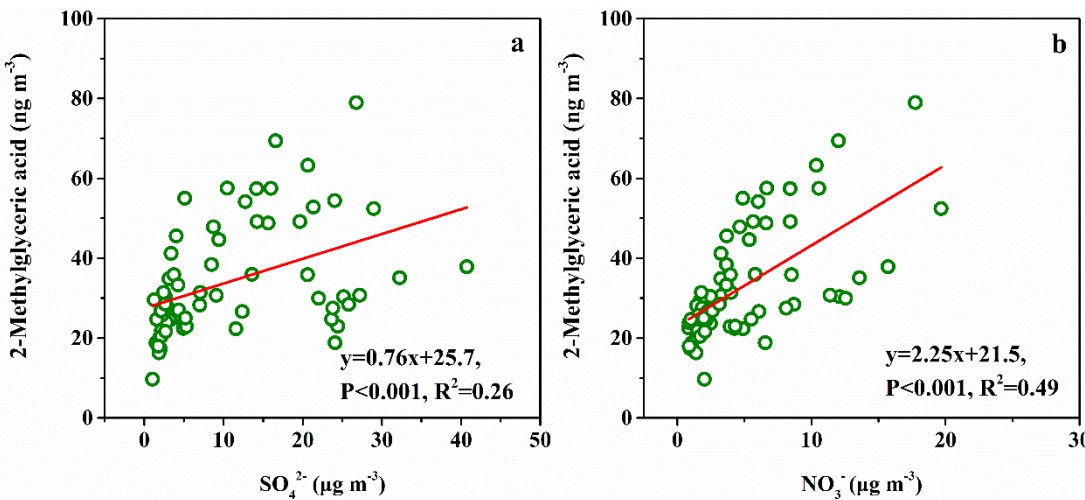

**Fig. 8.** Concentration correlation between (a) 2-methylglyceric acid (2-MGA) and $SO_4^{2-}$, (b) 2-
methylglyceric acid and $NO_3^-$ in the aerosols from Bode, Kathmandu.

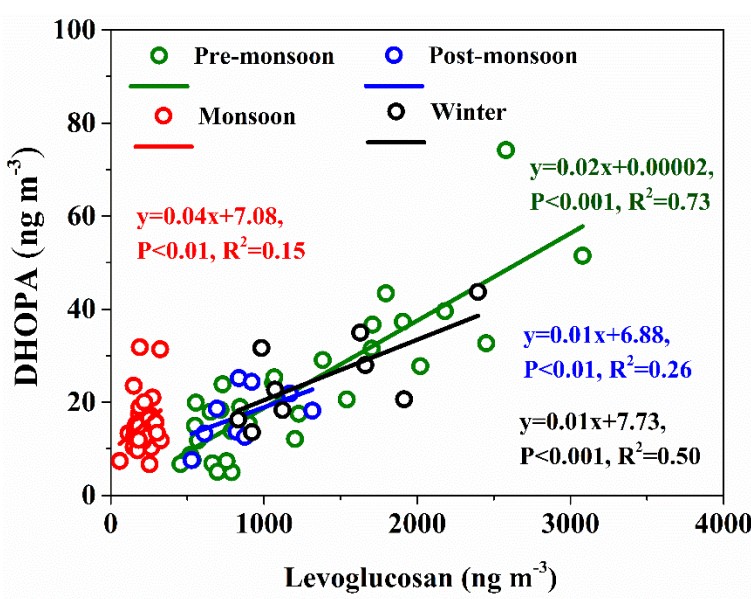

Fig. 9. Correlation between 2,3-dihydroxy-4-oxopentanoic acid (DHOPA) and levoglucosan in Bode

aerosols during the sampling period (April 2013 to April 2014).

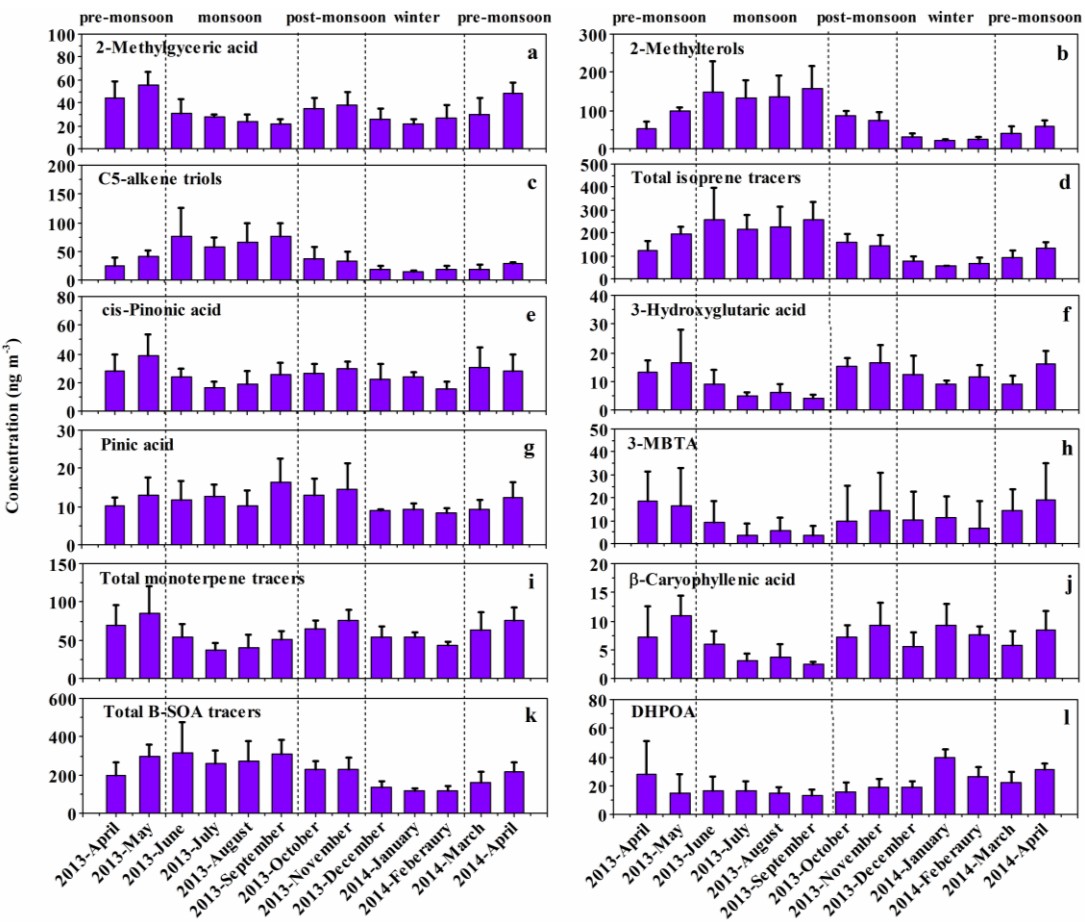

1083

**Fig. 10.** Monthly concentration variations of (a) BB-OC, (b) plant-debris-OC, (c) fungal-spore-OC, (d) SOC-isoprene (I-SOC), (e) SOC-monoterpenes (M-SOC), (f) SOC-sequiterpene (SOC-C), (g) SOC-isoprene+monoterpenes+sequiterpene (B-SOC), (h) SOC-toluene (A-SOC), and (i) total SOC that were estimated using a tracer-based method at Bode site, Kathmandu Valley during April 2013-April 2014.

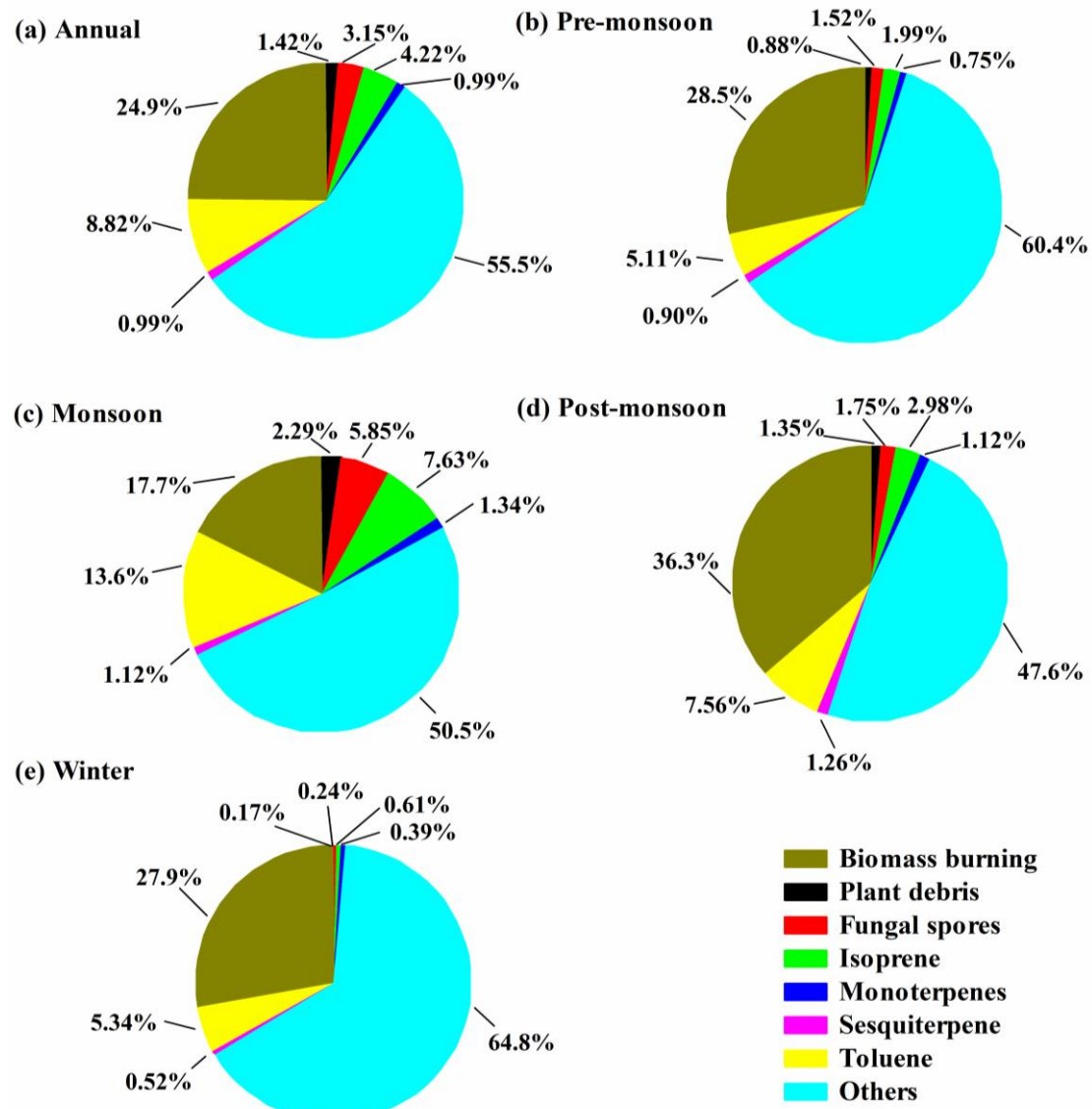

1088

**Fig. 11.** Pie-charts showing contributions from different sources to organic carbon based on the estimation of tracer-method in Bode, Kathmandu Valley: (a) annual, (b) pre-monsoon, (c) monsoon, (d) post-monsoon and (e) winter.