# Peer review of "Molecular characterization of organic aerosols in the Kathmandu Valley, Nepal: insights into primary and secondary sources"

_Atmospheric Chemistry and Physics, 2018_

## Referee Comment (RC1) · Anonymous Referee #1 · 2 Nov 2018

General comments This article presents a sizable set of field measurements for total suspended particulates in Bode, focusing on major inorganic ions and a wide range of markers to discuss primary and secondary emission sources. This work adopts organic molecular markers established in the published literature with tremendous effort given to chemical speciation which is commendable. In addition to providing data beneficial for control strategies and relevant implications, more clearly specified scientific novelty (new scientific findings that research communities have not known yet, and need to learn) will further enhance the value of this work. The major revision for the manuscript lies in the sampling artifacts and relevant impacts on the reported concentrations as well as discussion. Quartz filters are used for sample collection, which is recognized

to incur positive sampling artifacts by 10–20% for OC and up to 16 % for organic trac-
ers. The positive sampling artifacts on organics are cited based on a published study
(Ding et al. 2013) instead of experimental measurements devoted for this manuscript.
Authors are encouraged to examine appropriate ways (available in published literature)
to assess such positive artifacts and corresponding impacts on reported data, followed
by correction accordingly. At least, correction based on make-up experiments or post
data analyses need to be considered. Similar correction/discussion should be given to
the effects of discounted recovery rates. Quartz filters are also well known to adsorb
semi-volatile inorganics (e.g. nitrate, chloride, and ammonium), another type of major
positive sampling artifacts. How such effects affect the various correlations and discus-
sion involving inorganic ions mentioned in the manuscript deserve to be examined to
revise the discussion accordingly.

Specific comments Lns 118-120: Why would comparing only BC and O3 between Bode
and Paknajol be sufficient to conclude that Bode is a representative site for Kathmandu
Valley?

Ln 252: Incense burning can also emit levoglucosan. Would such emissions be signif-
icant at the study site?

Lns 450-451: Reference is needed for "During the fires, substantial amounts of
aerosols and VOCs including isoprene and monoterpenes would generate, . . .". Similar
description also appears at other locations.

Lns 452-453: An R2 value of 0.32 does not indicate good linear correlation between
Levoglucosan and 3-HGA, even though the p-value is less than 0.001. There are
also inconsistent use of "r" vs. "R2" throughout the work. The use of statistical mean
deserves more careful consideration and application.

Line 489-490: Typically, atmospheric samples show greatly fluctuated concentrations.
Prior to calculating and using the mean values for various comparison, a distribution
of concentrations can be examined to evaluate whether a median or mean should be

used to convene corresponding discussion.

Use of Lev/OC ratio among major biomass types deserves re-consideration. This use assumes that atmospheric degradation pattern of levoglucosan and overall OC at any time remains the same. This assumption is questionable, especially under varied temperature, relative humidity, locations, types and abundance of major biomass burnt, dominant burning conditions, varied transport, etc. It is also worth noting that oxidation intermediates of levoglucosan (and other organics) remain part of overall OC, which adds additional questions on the validity of adopting the ratio. The ratio at a given time point is a net results of multiple atmospheric processes on overall OC and levoglucosan therein.

---

## Referee Comment (RC2) · Anonymous Referee #2 · 12 Nov 2018

This manuscript presents measurement results from a year-long campaign at a site in the Himalayan-Tibetan Plateau. Ambient aerosol samples were subjected to chemical speciation, including molecular source tracers. Various methods were applied to identify emission sources and estimate the contributios from the individual sources. Primary and secondary organic aerosol components were assessed in the ambient aerosol, and the single largest source contribution was determined to be from biomass burning activities.

The results presented here are important for better understanding the properties and emission sources of organic aerosols at such critical sites as the Himalayan-Tibetan

[Figure]

Plateau, which have a profound influence on regional and even global climate. The paper, therefore, fits well within the scope of the journal, and is based on an extensive data set with adequate interpretation and discussions of the findings. Thus, I recommend publication of the manuscript in ACP, upon consideration of the comments and suggestions listed below.

Specific comments:

1. Lines 112-120: In the site description there is no mention of the airport (from which the met data were obtained, as stated in line 178) that is apparently in close proximity to the sampling site, and thus could have specific source influence on the collected samples. Please, add a brief statement regarding this potential impact, including the predominant wind patterns, i.e., during which periods the site is upwind and downwind of the airport.

2. Lines 264-166: The data plotted in this figure are apparently annual average values. It may be interesting to see the seasonal average numbers as well.

3. Lines 447-450: Do the authors have a possible explanation for the association of biomass burning emissions with SOA formation from monoterpenes? Is there a predominance of coniferous trees in the area which might have been subject to burning?

4. Lines 467-488: This statement should be made with caution, as a good correlation may also be due to other dominant source emissions which coincided with the biomass burning emissions.

5. Lines 471-473: An additional source of the uncertainties is the lack of representative source profiles for the given study location.

6. Line 504: Why would PBAP have a large contribution to the ambient PM at the sampling site?

7. Lines 514-517: It would be interesting to see a comparison here with measurements from other sites, reported in the literature.

[Figure]

**[ACPD](ACPD)**

Interactive
comment

8. Lines 549-551: If the authors mention dicarboxylic acids (DCAs) as an additional OC fraction, this implies that they are not associated with any of the sources for which estimates were made. What other sources would the DCAs be derived from?

Technical corrections:

1. Line 60: Omit "badly" before "poor".

2. Line 66: Delete the indefinite article "a" before "concern".

3. Lines 103 and 109: Delete the definite article "the" before "Kathmandu".

4. Lines 108 and 109: Add the definite article "the" before "central-eastern", "Nagarkot", and "Bode".

5. Line 117: Add "of" before "a mix"

6. Line 147: The sentence should start with "A trace gas chromatograph", and the name of the manufacturer is "Thermo Scientific".

7. Line 153: The first part of the sentence is not complete and therefore needs to be reworded; especially the word "While" is not fitting here.

8. Lines 160, 181, and 188: Add the definite article "the" before "current", "wet", and "Bode".

9. Line 162: Delete the definite article "the" before "artifacts"

10. Line 188: Change "are" to "is".

11. Line 190: Add "tracers" or "products" at the end of the sentence.

12. Line 202: Add "were observed" at the end of the sentence.

13. Lines 224-227: Revise the sentence as follows: "This is consistent with the seasonal variation of the precursors NOx, NO2 and SO2, which are mainly caused by automobile exhaust, household cooking, and operation of the typical biomass co-fired
brick kilns ..."

14. Line 228: Change the sentence to "... run on the Kathmandu Valley roads ..."

15. Lines 229-231: Revise the sentence as follows: "Diesel- or gasoline-powered generators (producing higher NOx emissions) and garbage burning are other major sources ..."

16. Lines 256, 270 and 272: Change "ranged" to "ranging".

17. Lines 270 and 271: Add "an" before "average".

18. Line 281: Add a comma after "pollen".

19. Lines 295, 398, and 559: Add "being" after "while".

20. Line 304: Change "complicated" to "complex".

21. Lines 326-327: Revise the sentence as follows: "In addition, the higher temperatures (Fig. S1a) were conducive for more active microbial activities."

22. Line 357: Use consistent terms for anhydrosugars, i.e., change "dehydrated sugars" to "anhydrosugars".

23. Line 376: Change "are" to "occurs".

24. Line 455: Delete "in".

25. Line 463: Change "the" to "a".

26. Lines 545-548: These sentences need to be polished.

27. Line 574: Change "show" to "shows".
* * *

---

## Author Comment (AC1) · 2 Dec 2018

We would like to thank the anonymous reviewer's helpful comments and suggestions which, we believe, have supported to improve the quality of the current manuscript. We have tried our best to incorporate the reviewers' comments in the manuscript. In the following responses, the reviewer' original comments are in black, authors' responses in blue and changes in the manuscript in red.

**Responses to Referee #1**

General comments:

This article presents a sizable set of field measurements for total suspended particulates in Bode, focusing on major inorganic ions and a wide range of markers to discuss primary and secondary emission sources. This work adopts organic molecular markers established in the published literature with tremendous effort given to chemical speciation which is commendable.

1. In addition to providing data beneficial for control strategies and relevant implications, more clearly specified scientific novelty (new scientific findings that research communities have not known yet, and need to learn) will further enhance the value of this work.

**Response:** Thanks for the constructive suggestion. We now added more descriptions of the scientific novelty in lines 634-633, "In addition, the current study based on the molecular level-source apportionment of OC in heavy polluted region from South Asia provides a much more specific quantification of source estimation for OC, which is different from previous studies based on the bulk carbonaceous aerosol using radiocarbon ($^{14}$C) measurements, PMF and CBM".

**2.** The major revision for the manuscript lies in the sampling artifacts and relevant impacts on the reported concentrations as well as discussion. Quartz filters are used for sample collection, which is recognized to incur positive sampling artifacts by 10–20% for OC and up to 16 % for organic tracers. The positive sampling artifacts on organics are cited based on a published study (Ding et al. 2013) instead of

experimental measurements devoted for this manuscript. Authors are encouraged to examine appropriate ways (available in published literature) to assess such positive artifacts and corresponding impacts on reported data, followed by correction accordingly. At least, correction based on make-up experiments or post data analyses need to be considered. Similar correction/discussion should be given to the effects of discounted recovery rates.

**Response:** We understand that the main concern is about the sampling artefacts. As far as we are aware, filter sampling using a high-volume sampler is a common method to collect atmospheric particles. There are mainly two types of filters. One is fiber filter (e.g., glass, quartz), the other is porous membrane filter (e.g., Teflon). We agree with the reviewer that a positive artifact may occur during sampling due to adsorption of gaseous species on the surface of quartz fiber filters. Alternatively, a negative sampling artefact may occur during sampling due to a loss of semi-volatile organic compounds from the aerosols collected on quartz fiber filters. Both evaporation and adsorption can be affected by changing pressure or temperature.

There are some studies trying to elaborate the positive and negative artifacts using the backup filter and denuder (Genberg et al., 2011;Subramanian et al., 2004;Yttri et al., 2011b;Yttri et al., 2011a;Gelencsér et al., 2007;McDow and Huntzicker, 1990;Chow et al., 2010;Cheng et al., 2009;Turpin et al., 2000). Subramanian et al. (2004) quantified the negative artifact to be small, typically less than 10% (6.3%±6.2%) of the OC by the denuded quartz filter with a carbon-impregnated glass fiber backup filter and the positive artifact of 10–20% according to the quartz behind quartz approach, respectively for the 24 h aerosol samples from a hill in Pittsburgh, Pennsylvania. Yttri et al. (2011a) reported the mean positive sampling artifact of OC ranged from 11±2 % at the Finnish site Hyytiälä to 18%± 4% at the Birkenes site in Norway. Cheng et al. (2009) reported 10% of the OC captured by the bare quartz filter was due to the positive artifact in Beijing, China, from January to February 2009.

Similar to bulk OC, the individual organic tracers also suffer from the effect of

sampling artifact. However, to our best, we did not find such detailed information in the previous literatures. Furthermore, the sampling artifacts differ from approaches, study regions and sampling period. Therefore, it is difficult for our current study to estimate the artifacts and make correction, which need a systematic and comprehensive study in the future in Kathmandu Valley and South Asia.

We reorganized the sentences denoting possible artifacts as "There may be positive and negative artifacts during the sample handling/conditioning due to the adsorption/evaporation processes of organic aerosols (Fu et al., 2010;Li et al., 2018;Boreddy et al., 2017;Oanh et al., 2016). In a comparable study, Ding et al. (2013) reported the positive artifacts of 10−20% for OC and up to 16% for organic tracers using a backup quartz filter placed behind the main quartz filter" in lines 132-136.

We also add the description about results of OC and molecular tracers in the field blank filters in lines 145-146 with "The concentrations of OC and EC from field blank filters were $0.59\pm0.13$ μg m$^{-3}$ and $0.00$ μg m$^{-3}$, respectively. The OC data reported here were blank corrected" and lines 169-170 of "Field blank filters were analyzed by the procedure used by the samples above, but no target compounds were detected."

The reviewer also suggested us to consider the effects by the discounted recovery rates. Regarding this point, Stone et al. (2012) developed an empirical approach to estimate the error from surrogate quantification (EQ) based on homologous series of atmospherically relevant compounds and applied that to the study in another rural site in Kathmandu. According to the method, now we also add the "estimation of measurement uncertainty" to our MS in Section 2.4 (Line 180-197) as "Since there is no commercial standard available for most SOA tracers (except for cis-pinonic acid and pinic acid), the use of surrogate standards for quantification introduces additional error to the measurements. Error in analyte measurement (EA) is propagated from the standard deviation of the field blank (EFB), error in spike recovery (ER) and the error from surrogate quantification (EQ):

$$EA = \sqrt{EFB^2 + ER^2 + EQ^2}$$

EFB was 0 in this study due to SOA tracers that were not detected in the field

blanks. The spike recoveries of surrogate standards were used to estimate the ER of tracers, ranging from 9.2% (erythritol) to 26.1% (cis-pinonic acid). According to Stone et al. (2012), there is an empirical approach to estimate EQ based on homologous series of atmospherically relevant compounds. The relative error introduced by each carbon atom (En) was estimated to be 15 %, each oxygenated functional group (Ef) to be 10% and alkenes (Ed) to be 60%. Therefore, the EQ are calculated as:

$$EQ=En\Delta n+ Ef\Delta f+ Ed\Delta d$$

where $\Delta n$, $\Delta f$ and $\Delta d$ are the difference of carbon atom number, oxygen-containing functional group and alkene functionality between a surrogate and an analyte, respectively.

The estimated uncertainties in tracer measurement is presented in Table S2. The EQ ranged from 15% (2-methyltetrols) to 120% (β-caryophyllenic acid) in this study. Propagated with the error in recovery, EA were estimated in the range of 17.6% to 122.4%."

Table S2 Estimation of measurement uncertainty

| Tracers | Tracer formula | Surrogates | Surrogate formula | EQ (%) | [a] ER (%) | EA (%) |
|---|---|---|---|---|---|---|
| *cis*-Pinonic acid | $C_{10}H_{16}O_3$ | *cis*-Pinonic acid | | | 26.1 | |
| Pinic acid | $C_9H_{14}O_4$ | Pinic acid | | | 23.9 | |
| 3-Methyl-1,2,3-butantricarboxylic acid | $C_8H_{12}O_6$ | *cis*-Pinonic acid | $C_{10}H_{16}O_3$ | 60 | 26.1 | 65.4 |
| 3-Hydroxyglutaric acid | $C_5H_8O_5$ | *cis*-Pinonic acid | $C_{10}H_{16}O_3$ | 95 | 26.1 | 98.5 |
| 3-Hydroxy-4,4-dimethylglutaric acid | $C_7H_{12}O_5$ | *cis*-Pinonic acid | $C_{10}H_{16}O_3$ | 65 | 26.1 | 70.0 |
| *cis*-2-Methyl-1,3,4-trihydroxy-1-butene | $C_5H_{10}O_3$ | Erythritol | $C_4H_{10}O_4$ | 85 | 9.2 | 85.5 |
| 3-Methyl-2,3,4-trihydroxy-1-butene | $C_5H_{10}O_3$ | Erythritol | $C_4H_{10}O_4$ | 85 | 9.2 | 85.5 |
| *trans*-2-Methyl-1,3,4-trihydroxy-1-butene | $C_5H_{10}O_3$ | Erythritol | $C_4H_{10}O_4$ | 85 | 9.2 | 85.5 |
| 2-Methylglyceric acid | $C_4H_8O_4$ | Erythritol | $C_4H_{10}O_4$ | 20 | 9.2 | 22.0 |
| 2-Methylthreitol | $C_5H_{12}O_4$ | Erythritol | $C_4H_{10}O_4$ | 15 | 9.2 | 17.6 |
| 2-Methylerythritol | $C_5H_{12}O_4$ | Erythritol | $C_4H_{10}O_4$ | 15 | 9.2 | 17.6 |
| β-Caryophyllenic acid | $C_{13}H_{20}O_4$ | Pinic acid | $C_9H_{14}O_4$ | 120 | 23.9 | 122.4 |
| 2,3-Dihydroxy-4-oxopentanoic acid | $C_5H_8O_5$ | Azelaic acid | $C_9H_{16}O_4$ | 90 | 12.8 | 90.9 |

[a] ER is the difference between 100% and mean recovery of each surrogate standard.

**2.** Quartz filters are also well known to adsorb semi-volatile inorganics (e.g. nitrate,

chloride, and ammonium), another type of major positive sampling artifact. How such effects affect the various correlations and discussion involving inorganic ions mentioned in the manuscript deserve to be examined to revise the discussion accordingly.

**Response:** We agree with the reviewer. Both the positive and negative artifacts may occur during sampling aerosols on the quartz filters. Single filter-based sampling and filter pack systems without any denuders or without backup filters are still widely used and the extent of the sampling artifacts of volatile species in these sampling systems is not well understood. Wei et al. (2015) reported the loss of $NH_4^+$, $NO_3^-$, and $Cl^-$ accounting for particulate matter, which ranged from 1.85% to 41.44% with a typical value of about 10%. Liu et al. (2014) showed that during 24 h sampling with denuder sampler at National Chiao-Tung University campus, Taiwan, the positive artifact of $NH_4^+$ and $Cl^-$ was not important for aerosol mass concentration, and existed in $NO_3^-$ species only, which was 5.0%±6.5% of actual $NO_3^-$ concentration. Timonen et al. (2014) reported a positive artifact of 1.3% ± 1.8% for ammonium and 42% ± 33% for nitrate of the $PM_1$ samples with back-up filters from an urban, background area near Helsinki city.

During our sampling, we used the single filter-based sampling and filter pack systems without any denuders or backup filters. Therefore, we cannot quantify the positive sampling artifacts of nitrate, chloride, and ammonium. If we used some correction ratios adopted from the previous studies, it will systematically modify the concentration data for those compounds; however, it will not affect the correlations among different compounds. Therefore, we intend to keep the current dataset without correction for the sampling artifact. Definitely, in the future study, we will choose more suitable sampler to reveal the effects of such sampling artifacts.

In addition, the concentrations of major ions reported in the MS have already been blank corrected. We add the description about results of major ions in lines 141-142 of "They denoted less than 5% of the real sample concentrations in the field blank filters (Tripathee et al., 2017)".

**Specific comments:**

**1.** Line118-120: Why would comparing only BC and $O_3$ between Bode and Paknajol be sufficient to conclude that Bode is a representative site for Kathmandu Valley?

**Response:** Actually, the descriptions from lines 103-122 are all about the explanation of Bode as a representative site for the Kathmandu Valley. More specifically, the BC and $O_3$ between Bode and Paknajol (in lines 118-120) are chosen as example to illustrate this claim.

**2.** Line 252: Incense burning can also emit levoglucosan. Would such emissions be significant at the study site?

**Response:** Yes, most incenses are made of wood powder (https://en.wikipedia.org), which can emit levoglucosan when they are burnt. There are religious activities in the Kathmandu Valley, so Bode may be influenced by the incense burning. However, we don't know how large its effect is. Now we add a sentence in lines 308-309 as "We must point out that the incense burning in Kathmandu Valley may also influence the levoglucosan concentration".

**3.** Line 450-451: Reference is needed for "During the fires, substantial amounts of aerosols and VOCs including isoprene and monoterpenes would generate, ...". Similar description also appears at other locations.

**Response:** Missing citation has been included in the reference. Please see lines 405 and 476.

**4.** Line 452-453: An $R^2$ value of 0.32 does not indicate good linear correlation between levoglucosan and 3-HGA, even though the p-value is less than 0.001. There is also inconsistent use of "r" vs. "$R^2$" throughout the work. The use of statistical mean deserves more careful consideration and application.

**Response:** Suggestion taken. All r were changed into $R^2$. Please see lines 257-259 and lines 320-322. Though the linear correlation coefficient is not very good, it indicates to some degree that monoterpene tracers may be influenced by biomass burning.

**5.** Line 489-490: Typically, atmospheric samples show greatly fluctuated concentrations. Prior to calculating and using the mean values for various comparison, a distribution of concentrations can be examined to evaluate whether a median or mean should be used to convene corresponding discussion. Use of Lev/OC ratio among major biomass types deserves re-consideration. This use assumes that atmospheric degradation pattern of levoglucosan and overall OC at any time remains the same. This assumption is questionable, especially under varied temperature, relative humidity, locations, types and abundance of major biomass burnt, dominant burning conditions, varied transport, etc. It is also worth noting that oxidation intermediates of levoglucosan (and other organics) remain part of overall OC, which adds additional questions on the validity of adopting the ratio. The ratio at a given time point is a net result of multiple atmospheric processes on overall OC and levoglucosan therein.

**Response:** Now we add the median concentrations in Table 1.

We totally agree with the referee that levoglucosan/OC (Lev/OC) ratios varied depending on biomass burning sources and conditions and degradation. The degradation of levoglucosan is affected by radicals (OH), temperature, and relative humidity (Hoffmann et al., 2010;Bai et al., 2013;Lai et al., 2014;Slade and Knopf, 2014). However, given the complicated biomass burning sources, conditions and degradation mechanism, it is not applicable to estimate the uncertainty for the moment.

Still, the Lev/OC ratio of ~8.2% in the burning source have been widely used (Graham et al., 2002;Fu et al., 2014;Ho et al., 2014;Sang et al., 2011;Zhu et al., 2016;Mkoma et al., 2013), especially in Asia. Although the ratios in the BB source emissions vary among different types of biomass fuels and burning conditions (Mochida et al., 2010). In this work, the Kathmandu valley is considered as a source region of organic aerosols, therefore, we believe that using Lev/OC ratio of 8.14% is reliable to estimate biomass burning contributions. The estimation can also be compared to other studies using the same ratio. Now we estimate the uncertainties

using different ratios from other studies in Table S3.

Table S3 Uncertainties using different ratios from other studies for biomass burning estimation

| | | Lev/OC ratios | | | | | |
|---|---|---|---|---|---|---|---|
| | | 8.14% | 8.27% | 7.94% | 14.0% | 12.0% | 10.1% |
| Pre-monsoon | Average | 28.5 | 28.0 | 29.2 | 16.6 | 19.3 | 23.0 |
| | Stdev | 10.3 | 10.1 | 10.5 | 5.96 | 6.96 | 8.29 |
| | Median | 28.0 | 27.5 | 28.7 | 16.3 | 19.0 | 22.6 |
| Monsoon | Average | 17.7 | 17.4 | 18.2 | 10.3 | 12.0 | 14.3 |
| | Stdev | 5.11 | 5.03 | 5.24 | 2.97 | 3.47 | 4.13 |
| | Median | 17.2 | 16.9 | 17.6 | 9.99 | 11.7 | 13.9 |
| Post-monsoon | Average | 36.3 | 35.8 | 37.3 | 21.1 | 24.7 | 29.4 |
| | Stdev | 10.4 | 10.3 | 10.7 | 6.07 | 7.08 | 8.44 |
| | Median | 32.3 | 31.8 | 33.2 | 18.8 | 21.9 | 26.1 |
| Winter | Average | 27.9 | 27.5 | 28.6 | 16.2 | 18.9 | 22.6 |
| | Stdev | 8.63 | 8.50 | 8.85 | 5.02 | 5.86 | 6.98 |
| | Median | 24.9 | 24.5 | 25.5 | 14.5 | 16.9 | 20.1 |
| Annual | Average | 24.9 | 24.6 | 25.6 | 14.5 | 16.9 | 20.2 |
| | Stdev | 10.4 | 10.3 | 10.7 | 6.07 | 7.08 | 8.44 |
| | Median | 22.4 | 22.1 | 23.0 | 13.0 | 15.2 | 18.1 |

We added the sentence of "although the ratios in the BB source emissions vary among different types of biomass fuels and burning conditions (Mochida et al., 2010)" in lines 508-509 and "The mean value of Lev/OC value of biomass burning from main biomass types was 10.1%. In this study, we choose the mostly used values of 8.14% for biomass burning estimation (Graham et al., 2002;Fu et al., 2014;Ho et al., 2014;Sang et al., 2011;Zhu et al., 2016;Mkoma et al., 2013). In addition, we also calculated the uncertainties of using different ratios (see Table S3), the diagnostic ratios among molecular tracers and OC (e.g., Lev/OC) from direct emissions are critical for more precise results. It's meaningful to understand the emission characteristics for individual OC emission categories, as well as in different locations, especially in South Asia." in lines 518-524. We reorganized the precaution as the second reviewer suggested in lines 501-503, which is "It should be noted here that tracer methods can provide a reasonable estimation, but uncertainties are introduced considering the site differences and the lack of representative source profiles for the

given study location. The contribution evaluated from each source to OC in the current study is still inferable".

**Reference**

Bai, J., Sun, X. M., Zhang, C. X., Xu, Y. S., and Qi, C. S.: The OH-initiated atmospheric reaction mechanism and kinetics for levoglucosan emitted in biomass burning, Chemosphere, 93, 2004-2010, 10.1016/j.chemosphere.2013.07.021, 2013.

Boreddy, S. K. R., Kawamura, K., and Tachibana, E.: Long-term (2001-2013) observations of water-soluble dicarboxylic acids and related compounds over the western North Pacific: trends, seasonality and source apportionment, Scientific Reports, 7, 10.1038/s41598-017-08745-w, 2017.

Cheng, Y., He, K. B., Duan, F. K., Zheng, M., Ma, Y. L., and Tan, J. H.: Positive sampling artifact of carbonaceous aerosols and its influence on the thermal-optical split of OC/EC, Atmospheric Chemistry and Physics, 9, 7243-7256, 10.5194/acp-9-7243-2009, 2009.

Chow, J. C., Watson, J. G., Chen, L. W. A., Rice, J., and Frank, N. H.: Quantification of PM2.5 organic carbon sampling artifacts in US networks, Atmospheric Chemistry and Physics, 10, 5223-5239, 10.5194/acp-10-5223-2010, 2010.

Ding, X., Wang, X., Xie, Z., Zhang, Z., and Sun, L.: Impacts of Siberian Biomass Burning on Organic Aerosols over the North Pacific Ocean and the Arctic: Primary and Secondary Organic Tracers, Environmental Science & Technology, 47, 3149-3157, 10.1021/es3037093, 2013.

Fu, P., Kawamura, K., Chen, J., and Miyazaki, Y.: Secondary production of organic aerosols from biogenic VOCs over Mt. Fuji, Japan, Environmental Science & Technology, 48, 8491-8497, 2014.

Fu, P. Q., Kawamura, K., Pavuluri, C. M., Swaminathan, T., and Chen, J.: Molecular characterization of urban organic aerosol in tropical India: contributions of primary emissions and secondary photooxidation, Atmospheric Chemistry and Physics, 10, 2663-2689, 2010.

Gelencsér, A., May, B., Simpson, D., Sánchez-Ochoa, A., Kasper-Giebl, A., Puxbaum, H., Caseiro, A., Pio, C., and Legrand, M.: Source apportionment of PM2.5 organic aerosol over Europe: Primary/secondary, natural/anthropogenic, and fossil/biogenic origin, Journal of Geophysical Research: Atmospheres, 112, 10.1029/2006JD008094, 2007.

Genberg, J., Hyder, M., Stenström, K., Bergström, R., Simpson, D., Fors, E. O., Jönsson, J. Å., and Swietlicki, E.: Source apportionment of carbonaceous aerosol in southern Sweden, Atmospheric Chemistry and Physics, 11, 11387-11400, 10.5194/acp-11-11387-2011, 2011.

Graham, B., Mayol-Bracero, O. L., Guyon, P., Roberts, G. C., Decesari, S., Facchini, M. C., Artaxo, P., Maenhaut, W., Koll, P., and Andreae, M. O.: Water-soluble organic compounds in biomass burning aerosols over Amazonia - 1. Characterization by NMR and GC-MS, Journal of Geophysical Research-Atmospheres, 107, 10.1029/2001jd000336, 2002.

Ho, K. F., Engling, G., Ho, S. S. H., Huang, R., Lai, S., Cao, J., and Lee, S. C.: Seasonal variations of anhydrosugars in PM2.5 in the Pearl River Delta Region, China, Tellus Series B-Chemical and Physical Meteorology, 66, 10.3402/tellusb.v66.22577, 2014.

Hoffmann, D., Tilgner, A., Iinuma, Y., and Herrmann, H.: Atmospheric Stability of Levoglucosan: A Detailed Laboratory and Modeling Study, Environmental Science & Technology, 44, 694-699,

10.1021/es902476f, 2010.

Lai, C. Y., Liu, Y. C., Ma, J. Z., Ma, Q. X., and He, H.: Degradation kinetics of levoglucosan initiated by hydroxyl radical under different environmental conditions, Atmospheric Environment, 91, 32-39, 10.1016/j.atmosenv.2014.03.054, 2014.

Li, J., Wang, G., Wu, C., Cao, C., Ren, Y., Wang, J., Li, J., Cao, J., Zeng, L., and Zhu, T.: Characterization of isoprene-derived secondary organic aerosols at a rural site in North China Plain with implications for anthropogenic pollution effects, Scientific Reports, 8, 10.1038/s41598-017-18983-7, 2018.

Liu, C.-N., Lin, S.-F., Awasthi, A., Tsai, C.-J., Wu, Y.-C., and Chen, C.-F.: Sampling and conditioning artifacts of PM2.5 in filter-based samplers, Atmospheric Environment, 85, 48-53, 2014.

McDow, S. R., and Huntzicker, J. J.: Vapor adsorption artifact in the sampling of organic aerosol: Face velocity effects, Atmospheric Environment. Part A. General Topics, 24, 2563-2571, 1990.

Mkoma, S. L., Kawamura, K., and Fu, P. Q.: Contributions of biomass/biofuel burning to organic aerosols and particulate matter in Tanzania, East Africa, based on analyses of ionic species, organic and elemental carbon, levoglucosan and mannosan, Atmospheric Chemistry and Physics, 13, 10325-10338, 10.5194/acp-13-10325-2013, 2013.

Mochida, M., Kawamura, K., Fu, P. Q., and Takemura, T.: Seasonal variation of levoglucosan in aerosols over the western North Pacific and its assessment as a biomass-burning tracer, Atmospheric Environment, 44, 3511-3518, 10.1016/j.atmosenv.2010.06.017, 2010.

Oanh, N. T. K., Hang, N. T., Aungsiri, T., Worrarat, T., and Danutawat, T.: Characterization of Particulate Matter Measured at Remote Forest Site in Relation to Local and Distant Contributing Sources, Aerosol and Air Quality Research, 16, 2671-2684, 10.4209/aaqr.2015.12.0677, 2016.

Sang, X.-F., Chan, C.-Y., Engling, G., Chan, L.-Y., Wang, X.-M., Zhang, Y.-N., Shi, S., Zhang, Z.-S., Zhang, T., and Hu, M.: Levoglucosan enhancement in ambient aerosol during springtime transport events of biomass burning smoke to Southeast China, Tellus Series B-Chemical and Physical Meteorology, 63, 129-139, 10.1111/j.1600-0889.2010.00515.x, 2011.

Slade, J. H., and Knopf, D. A.: Multiphase OH oxidation kinetics of organic aerosol: The role of particle phase state and relative humidity, Geophysical Research Letters, 41, 5297-5306, 10.1002/2014gl060582, 2014.

Stone, E. A., Nguyen, T. T., Pradhan, B. B., and Dangol, P. M.: Assessment of biogenic secondary organic aerosol in the Himalayas, Environmental Chemistry, 9, 263-272, 10.1071/en12002, 2012.

Subramanian, R., Khlystov, A. Y., Cabada, J. C., and Robinson, A. L.: Positive and Negative Artifacts in Particulate Organic Carbon Measurements with Denuded and Undenuded Sampler Configurations Special Issue of Aerosol Science and Technology on Findings from the Fine Particulate Matter Supersites Program, Aerosol Science and Technology, 38, 27-48, 10.1080/02786820390229354, 2004.

Timonen, H., Aurela, M., Carbone, S., Saarnio, K., Frey, A., Saarikoski, S., Teinila, K., Kulmala, M., and Hillamo, R.: Seasonal and diurnal changes in inorganic ions, carbonaceous matter and mass in ambient aerosol particles in an urban, background area, Boreal Environment Research, 19, 71-86, 2014.

Tripathee, L., Kang, S., Rupakheti, D., Cong, Z., Zhang, Q., and Huang, J.: Chemical characteristics of soluble aerosols over the central Himalayas: insights into spatiotemporal variations and

sources, Environmental Science and Pollution Research, 24, 24454-24472, 10.1007/s11356-017-0077-0, 2017.

Turpin, B. J., Saxena, P., and Andrews, E.: Measuring and simulating particulate organics in the atmosphere: problems and prospects, Atmospheric Environment, 34, 2983-3013, 2000.

Wei, L., Duan, J., Tan, J., Ma, Y., He, K., Wang, S., Huang, X., and Zhang, Y.: Gas-to-particle conversion of atmospheric ammonia and sampling artifacts of ammonium in spring of Beijing, Science China Earth Sciences, 58, 345-355, 10.1007/s11430-014-4986-1, 2015.

Yttri, K. E., Simpson, D., Nøjgaard, J. K., Kristensen, K., Genberg, J., Stenström, K., Swietlicki, E., Hillamo, R., Aurela, M., Bauer, H., Offenberg, J. H., Jaoui, M., Dye, C., Eckhardt, S., Burkhart, J. F., Stohl, A., and Glasius, M.: Source apportionment of the summer time carbonaceous aerosol at Nordic rural background sites, Atmospheric Chemistry and Physics, 11, 13339-13357, 10.5194/acp-11-13339-2011, 2011a.

Yttri, K. E., Simpson, D., Stenström, K., Puxbaum, H., and Svendby, T.: Source apportionment of the carbonaceous aerosol in Norway – quantitative estimates based on [14]C, thermal-optical and organic tracer analysis, Atmospheric Chemistry and Physics, 11, 9375-9394, 10.5194/acp-11-9375-2011, 2011b.

Zhu, C., Kawamura, K., Fukuda, Y., Mochida, M., and Iwamoto, Y.: Fungal spores overwhelm biogenic organic aerosols in a midlatitudinal forest, Atmospheric Chemistry and Physics, 16, 7497-7506, 10.5194/acp-16-7497-2016, 2016.

---

## Author Comment (AC2) · 2 Dec 2018

We would like to thank the anonymous reviewer's helpful comments and suggestions which, we believe, have supported to improve the quality of the current manuscript. We have tried our best to incorporate the reviewer' comments in the manuscript. In the following responses, the reviewer' original comments are in black, authors' responses in blue and changes in the manuscript in red.

**Responses to Referee #2**

This manuscript presents measurement results from a year-long campaign at a site in the Himalayan-Tibetan Plateau. Ambient aerosol samples were subjected to chemical speciation, including molecular source tracers. Various methods were applied to identify emission sources and estimate the contributions from the individual sources. Primary and secondary organic aerosol components were assessed in the ambient aerosol, and the single largest source contribution was determined to be from biomass burning activities. The results presented here are important for better understanding the properties and emission sources of organic aerosols at such critical sites as the Himalayan-Tibetan Plateau, which have a profound influence on regional and even global climate. The paper, therefore, fits well within the scope of the journal, and is based on an extensive data set with adequate interpretation and discussions of the findings. Thus, I recommend publication of the manuscript in ACP, upon consideration of the comments and suggestions listed below.

**Response:** We thank the referee for the positive evaluation on our work. We have adopted most of the comments to improve the manuscript substantially.

**Specific comments:**

**1.** Lines 112-120: In the site description there is no mention of the airport (from which the met data were obtained, as stated in line 178) that is apparently in close proximity to the sampling site, and thus could have specific source influence on the collected samples. Please, add a brief statement regarding this potential impact, including the predominant wind patterns, i.e., during which periods the site is upwind and downwind

of the airport.

**Response:** We agree with the reviewer. The Tribhuvan international airport was located west of the site ($\sim$ 4 km from Bode). It may influence the organic aerosols via fossil fuel combustion. However, in our current study, we only report one toluene tracer, which cannot track the pollution from the airport. Regarding the wind patterns, as described in the Section 2.1, the local wind direction varied all the time during the whole day sampling, and the pollution sources are mixed. Thus, currently it is hard to reveal the effect of airport emission by analyzing the wind pattern. Now we add a sentence of "The Tribhuvan international airport in the west of Bode ($\sim$ 4 km) may have potential impacts when there is westerly wind" in lines 116-117.

**2.** Lines 264-166: The data plotted in this figure are apparently annual average values. It may be interesting to see the seasonal average numbers as well.

**Response:** We totally agree with the referee. However, due to the sampler breakdown, power interruption and maintaining, the sample distribution is not uniform, there were less samples during post-monsoon (sample number = 9) and winter (sample number = 9). In order to obtain the convinced relationship correlation, we choose to plot the figure using the annual average values.

**3.** Lines 447-450: Do the authors have a possible explanation for the association of biomass burning emissions with SOA formation from monoterpenes? Is there a predominance of coniferous trees in the area which might have been subject to burning?

**Response:** We now add the explanation as "The forests in the Kathmandu Valley consist of broad-leaved evergreen mixed forest of Schima castanopsis at the base, oak-laurel forest in the middle (1800 to 2400 m a. s. l.) and oak forest at the top, while the conifer tree species Pinus roxiburghii (Khote Salla) and Pinus wallichiana (Gobre Salla) are also found (Department of Plant Resources, 2015;Sarkar et al., 2016). Monoterpenes were chiefly emitted from needle leaf trees (coniferous trees) (Kang et al., 2018). The forests in the Kathmandu Valley consist of broad-leaved evergreen mixed forest of Schima castanopsis at the base, oak-laurel forest in the middle (1800

to 2400 m a. s. l.) and oak forest at the top, while the conifer tree species Pinus roxiburghii (Khote Salla) and Pinus wallichiana (Gobre Salla) are also found (Department of Plant Resources, 2015;Sarkar et al., 2016). Monoterpenes were chiefly emitted from needle leaf trees (coniferous trees) (Kang et al., 2018). Therefore, it suggested that biomass-burning activities have had a significant influence on the atmospheric composition over Kathmandu Valley, especially for SOA tracers" in lines 478-483.

**4.** Lines 467-488: This statement should be made with caution, as a good correlation may also be due to other dominant source emissions which coincided with the biomass burning emissions.

**Response:** What we want to do in lines 467-488 is to roughly discuss the influencing factors that can have an impact on the toluene- SOC concentration, thus shed light on further study to concentrate on the influencing factors concerning the SOA formation. Therefore, we use univariate analysis to see which factor may influence the apportioned SOC and see the correlation between the potential influencing factors and the apportioned SOC. The correlation between different parameters could at least enlighten us of the influencing factors for SOA formation in megacities such as Kathmandu under the complex air pollution conditions.

**5.** Lines 471-473: An additional source of the uncertainties is the lack of representative source profiles for the given study location.

**Response:** We changed the expression as "It should be noted here that tracer methods can provide a reasonable estimation, but uncertainties are introduced considering the site differences and the lack of representative source profiles for the given study location." Please check lines 502-503.

**6.** Line 504: Why would PBAP have a large contribution to the ambient PM at the sampling site?

**Response:** As discussed in Section 3.3.2 and 3.3.3, "Notably, the levels of PBAP discussed above were much higher than other sites in the world (Zhu et al., 2015;Chen

et al., 2013;Liang et al., 2016), indicating the strong fungal spore production in the Kathmandu Valley during the monsoon season". Therefore, for the contribution estimation, we infer that the PBAP may have a large contribution to the ambient PM at the study site.

**7.** Lines 514-517: It would be interesting to see a comparison here with measurements from other sites, reported in the literature.

**Response:** We appreciate for the referee's comments. We now add the comparison with measurements from other sites. "There are also some similar results from the literatures. For example, Zhu et al. (2016) reported the contribution of plant debris to OC was 5.6% in nighttime and 4.6% in daytime respectively from aerosols in a mid-latitudinal forest. Szidat et al. (2006) reported the plant debris contributed to 3.2% of OC during summer in urban aerosols collected in Zurich, Switzerland. Fungal-spore-derived OC was the biggest contributor to total OC of 3.1 % (0.03 %– 19.8 %) in marine aerosols collected over the East China Sea during 18 May to 12 June 2014 (Kang et al., 2018). The study in the aerosols of Brazil urban site showed the mean contributions of fungal aerosol to OC was 8% (Emygdio et al., 2018). Liang et al. (2017) reported the contributions of fungal spores to OC of 1.2 ± 0.7% and 3.5 ± 3.7% in aerosols from an urban site and a rural site respectively during an entire year in Beijing, China. All above strengthened the importance of plant-debris and fungal spores to the aerosol burden in the atmosphere. Please see lines 552-561.

**8.** Lines 549-551: If the authors mention dicarboxylic acids (DCAs) as an additional OC fraction, this implies that they are not associated with any of the sources for which estimates were made. What other sources would the DCAs be derived from?

**Response:** Dicarboxylic acids (DCAs) can be emitted both from primary and secondary sources (Kawamura and Bikkina, 2016). There may be POC and SOC contribution to DCAs. However, in the current study, we didn't detect and consider DCAs, and the contribution from DCAs is difficult to be quantified. Legrand et al. (2013) reported mono- and di-carboxylic acids, originating from a broad range of primary organic compounds, could contribute 38–44 % of OC. Therefore, the others

may include the contribution from the DCAs. We rephrased the sentence as "Additionally, low molecular weight (LMW) dicarboxylic acids from both primary and secondary sources also constitute a significant fraction of atmospheric organic aerosols (Kawamura and Bikkina, 2016)" in lines 593-595.

**Technical corrections:**

**1.** Line 60: Omit "badly" before "poor".

**Response:** Corrected. Please see line 60.

**2.** Line 66: Delete the indefinite article "a" before "concern".

**Response:** Corrected. Please see line 66.

**3.** Lines 103 and 109: Delete the definite article "the" before "Kathmandu".

**Response:** Corrected. Please see lines 103 and 109.

**4.** Lines 108 and 109: Add the definite article "the" before "central-eastern", "Nagarkot", and "Bode".

**Response:** Corrected. Please see lines 108 and 109

**5.** Line 117: Add "of" before "a mix".

**Response:** Added. Please see lines 118.

**6.** Line 147: The sentence should start with "A trace gas chromatograph", and the name of the manufacturer is "Thermo Scientific".

**Response:** Corrected. Please see lines 155-156 as "A trace gas chromatography coupled to a Polaris Q mass spectrometry detector (GC-MS, Thermo Scientific) was used for analysis."

**7.** Line 153: The first part of the sentence is not complete and therefore needs to be reworded; especially the word "While" is not fitting here.

**Response:** We reorganized the expression as "For quantitative analysis, calibration curves were established by using authentic standards that were processed as described above. For the quantification of target compounds that were no available standards, they were estimated by the following surrogate compounds:" in lines 161-162.

**8.** Lines 160, 181, and 188: Add the definite article "the" before "current", "wet", and "Bode".

**Response:** Corrected. Please see lines 169, 204 and 211.

**9.** Line 162: Delete the definite article "the" before "artifacts".

**Response:** Deleted. Please see line 133.

**10.** Line 188: Change "are" to "is".

**Response:** Corrected. Please see lines 211.

**11.** Line 190: Add "tracers" or "products" at the end of the sentence.

**Response:** Added. Please see line 213.

**12.** Line 202: Add "were observed" at the end of the sentence.

**Response:** Added. Please see line 225.

**13.** Lines 224-227: Revise the sentence as follows: "This is consistent with the seasonal variation of the precursors $NO_x$, $NO_2$ and $SO_2$, which are mainly caused by automobile exhaust, household cooking, and operation of the typical biomass co-fired brick kilns ...".

**Response:** Revised as you suggested. Please see line 248-249.

**14.** Line 228: Change the sentence to "... run on the Kathmandu Valley roads ...".

**Response:** Changed. Now the sentence is "Currently, nearly 50% of the total motor vehicles in Nepal (approximately 2.33 million) run within on the Kathmandu Valley roads". Please see line 251.

**15.** Lines 229-231: Revise the sentence as follows: "Diesel- or gasoline-powered generators (producing higher NOx emissions) and garbage burning are other major sources ...".

**Response:** Changed. Now the sentence is "Diesel- or gasoline-powered generators (producing higher NOx emissions) and garbage burning are other major sources of air pollution in Nepal during the sampling period, which can also emit many aerosol precursors". Please see line 252-253.

**16.** Lines 256, 270 and 272: Change "ranged" to "ranging".

**Response:** Changed. Please see lines 279, 293 and 295.

**17.** Lines 270 and 271: Add "an" before "average".

**Response:** Added. Please see lines 293 and 294.

**18.** Line 281: Add a comma after "pollen".

**Response:** Corrected. Please see line 313.

**19.** Lines 295, 398, and 559: Add "being" after "while".

**Response:** Corrected. Please see lines 319, 422, and we change "being" to "their concentrations were" in line 604.

**20.** Line 304: Change "complicated" to "complex".

**Response:** Changed. Please see line 328.

**21.** Lines 326-327: Revise the sentence as follows: "In addition, the higher temperatures (Fig. S1a) were conducive for more active microbial activities."

**Response:** We changed the sentence as you suggested. Please see lines 351.

**22.** Line 357: Use consistent terms for anhydrosugars, i.e., change "dehydrated sugars" to "anhydrosugars".

**Response:** Corrected. Please see line 381.

**23.** Line 376: Change "are" to "occurs".

**Response:** Changed. Please see lines 400.

**24.** Line 455: Delete "in".

**Response:** Deleted. Please see lines 479.

**25.** Line 463: Change "the" to "a".

**Response:** Changed. Please see line 492.

**26.** Lines 545-548: These sentences need to be polished.

**Response:** We polished the sentences as "Nevertheless, there is still part of OC (55.5%) that we were not able to be attributed to any specific sources based on the

tracers analyzed in current study. There are partly uncertainties caused by the organic tracer analyses (estimation of measurement uncertainty was shown in Table S2)". Please check lines 589-592.

**27.** Line 574: Change "show" to "shows".

**Response:** Changed. Please see line 618.

**Reference**

Chen, J., Kawamura, K., Liu, C.-Q., and Fu, P.: Long-term observations of saccharides in remote marine aerosols from the western North Pacific: A comparison between 1990-1993 and 2006-2009 periods, Atmospheric Environment, 67, 448-458, 10.1016/j.atmosenv.2012.11.014, 2013.

Department of Plant Resources, N.: Bulletin of Department of Plant Resources, Nepal 37, Ministry of Forests and Soil Conservation, Kathmandu, 1-122, 2015.

Emygdio, A. P. M., Andrade, M. d. F., Gonçalves, F. L. T., Engling, G., Zanetti, R. H. d. S., and Kumar, P.: Biomarkers as indicators of fungal biomass in the atmosphere of São Paulo, Brazil, Science of The Total Environment, 612, 809-821, 2018.

Kang, M., Fu, P., Kawamura, K., Yang, F., Zhang, H., Zang, Z., Ren, H., Ren, L., Zhao, Y., Sun, Y., and Wang, Z.: Characterization of biogenic primary and secondary organic aerosols in the marine atmosphere over the East China Sea, Atmospheric Chemistry and Physics, 18, 13947-13967, 10.5194/acp-18-13947-2018, 2018.

Kawamura, K., and Bikkina, S.: A review of dicarboxylic acids and related compounds in atmospheric aerosols: Molecular distributions, sources and transformation, Atmospheric Research, 170, 140-160, 2016.

Legrand, M., Preunkert, S., May, B., Guilhermet, J., Hoffman, H., and Wagenbach, D.: Major 20th century changes of the content and chemical speciation of organic carbon archived in Alpine ice cores: Implications for the long-term change of organic aerosol over Europe, Journal of Geophysical Research: Atmospheres, 118, 3879-3890, doi:10.1002/jgrd.50202, 2013.

Liang, L., Engling, G., Du, Z., Cheng, Y., Duan, F., Liu, X., and He, K.: Seasonal variations and source estimation of saccharides in atmospheric particulate matter in Beijing, China, Chemosphere, 150, 365-377, 10.1016/j.chemosphere.2016.02.002, 2016.

Liang, L., Engling, G., Du, Z., Duan, F., Cheng, Y., Liu, X., and He, K.: Contribution of fungal spores to organic carbon in ambient aerosols in Beijing, China, Atmospheric Pollution Research, 8, 351-358, 10.1016/j.apr.2016.10.007, 2017.

Sarkar, C., Sinha, V., Kumar, V., Rupakheti, M., Panday, A., Mahata, K. S., Rupakheti, D., Kathayat, B., and Lawrence, M. G.: Overview of VOC emissions and chemistry from PTR-TOF-MS measurements during the SusKat-ABC campaign: high acetaldehyde, isoprene and isocyanic acid in wintertime air of the Kathmandu Valley, Atmospheric Chemistry and Physics, 16, 3979-4003, 2016.

Szidat, S., Jenk, T. M., Synal, H.-A., Kalberer, M., Wacker, L., Hajdas, I., Kasper-Giebl, A., and Baltensperger, U.: Contributions of fossil fuel, biomass-burning, and biogenic emissions to carbonaceous aerosols in Zurich as traced by 14C, Journal of Geophysical Research:

Atmospheres, 111, doi:10.1029/2005JD006590, 2006.

Zhu, C., Kawamura, K., and Kunwar, B.: Organic tracers of primary biological aerosol particles at subtropical Okinawa Island in the western North Pacific Rim, Journal of Geophysical Research-Atmospheres, 120, 5504-5523, 10.1002/2015jd023611, 2015.

Zhu, C., Kawamura, K., Fukuda, Y., Mochida, M., and Iwamoto, Y.: Fungal spores overwhelm biogenic organic aerosols in a midlatitudinal forest, Atmospheric Chemistry and Physics, 16, 7497-7506, 10.5194/acp-16-7497-2016, 2016.

---

## Author Response (AR2)

**Response to Co-Editor's comments**

We would like to thank Co-Editor Prof. Kim Oanh for the constructive comments and suggestions to improve our manuscript. We have carefully considered the comments and the replies are listed below.

**Comments to the Author:**

1) The authors are encouraged to place points of improvements for future studies in the supporting information file. Some key recommendations for future studies should be included in the main text in section 4 (Summary and conclusions). This will demonstrate the authors' rigor research quality and benefit the ACP research community.

**Response:** We now added the improvements needed for future research in the supporting information file (lines 23-50 in SI) and the key recommendations for future studies in section 4 of the MS with "There are additional improvements for future studies to be addressed in the supporting information file. The key recommendations are as follows: (i) much more tracers need to be identified to explain the other sources of organic aerosols in the KV; (ii) the conversion factors of tracers to organic carbon from local emissions are critical for more precise source apportionments and therefore, a number of studies on the emission characteristics will be valuable; (iii) comprehensive methods (e.g., carbon isotope and modeling) need to be integrated together for the source apportionment of organic aerosols in the KV; (iv) the influences of BB on the formation of secondary organic aerosols could be further studied, especially during the heavily polluted dry season, with additional simultaneous measurements of precursors (e.g., NOx and $O_3$), $PM_{2.5}$ and so on at the same time; (v) to better understand the atmospheric processes of various chemical species, investigations of size-segregated aerosols are especially needed in the heavy polluted KV." (lines 620-630).

In addition, we have carefully checked the English for proper usage of grammar and syntax in the revised manuscript with red color font. The language editing which has been done by a native English speaker.

[revised manuscript text omitted]